# Rethinking External Slow-Thinking:
# From Snowball Errors to Probability of Correct Reasoning

**Zeyu Gan** [1 2 3]   **Yun Liao** [4]   **Yong Liu** [1 2 3]

## Abstract

Test-time scaling, which is also often referred to as *slow-thinking*, has been demonstrated to enhance multi-step reasoning in large language models (LLMs). However, despite its widespread utilization, the mechanisms underlying slow-thinking methods remain poorly understood. This paper explores the mechanisms of external slow-thinking from a theoretical standpoint. We begin by examining the snowball error effect within the LLM reasoning process and connect it to the likelihood of correct reasoning using information theory. Building on this, we show that external slow-thinking methods can be interpreted as strategies to mitigate the error probability. We further provide a comparative analysis of popular external slow-thinking approaches, ranging from simple to complex, highlighting their differences and interrelationships. Our findings suggest that the efficacy of these methods is not primarily determined by the specific framework employed, and that expanding the search scope or the model's internal reasoning capacity may yield more sustained improvements in the long term. We open-source our code at https://github.com/ZyGan1999/Snowball-Errors-and-Probability.

## 1. Introduction

Scaling laws (Kaplan et al., 2020) have been widely accepted as a guiding principle in the development of large language models (LLMs), indicating that the performance of LLMs improves with the growth of model size and training data. Over the past few years, the trend in this field has been toward expanding the scale of training phase, resulting in significant performance improvements (Yuan et al., 2023; Zelikman et al., 2022; Rafailov et al., 2024). However, the marginal gains in model performance diminish as scale increases, and training more powerful models necessitates a substantial rise in investment. Consequently, recent researches have shifted focus to scaling strategies beyond model size, including optimizations during the post-training phase and even at the test-time stage (Snell et al., 2024).

Following the release of LLMs with remarkable reasoning capabilities, such as OpenAI's o1 (2024), DeepSeek's R1 (2025), and Qwen's QwQ (2024b), it has become widely acknowledged that scaling the inference process of LLMs offers a promising avenue for further enhancing model performance. Specifically, empirical studies have shown that the reasoning quality of LLMs improves with extended inference time (Lightman et al., 2023). This observation has sparked a new research trajectory focused on augmenting the reasoning abilities of LLMs by increasing inference costs during the test-time phase, a concept referred to as *test-time scaling*, or more colloquially, *slow-thinking*.

Test-time scaling strategies can be generally classified into two primary approaches: *internal* and *external* slow-thinking (Jiang et al., 2024; Min et al., 2024). Internal slow-thinking involves adjusting model parameters through additional training on specifically designed reasoning tasks, aiming to inherently extend the model's output length and thereby enhance its reasoning capabilities. In contrast, external slow-thinking focuses on increasing inference costs by introducing additional computational steps, such as re-sampling or re-generating model outputs multiple times (Brown et al., 2024), thereby prolonging inference time and improving reasoning quality.

**This paper focuses on external slow-thinking techniques,** which are inspired by human cognitive processes. When facing complex questions, humans often take extra time to reflect and refine their intermediate answers, leading to greater accuracy. Similarly, external slow-thinking methods, such as the Best-of-N (BoN) strategy, draw multiple samples and evaluate them using techniques like majority voting

[1]Gaoling School of Artificial Intelligence, Renmin University of China, Beijing, China [2]Beijing Key Laboratory of Research on Large Models and Intelligent Governance [3]Engineering Research Center of Next-Generation Intelligent Search and Recommendation, MOE [4]College of Artificial Intelligence, Tianjin University of Science and Technology, Tianjin, China. Correspondence to: Yong Liu <liuyonggsai@ruc.edu.cn>.

*Proceedings of the 42nd International Conference on Machine Learning*, Vancouver, Canada. PMLR 267, 2025. Copyright 2025 by the author(s).

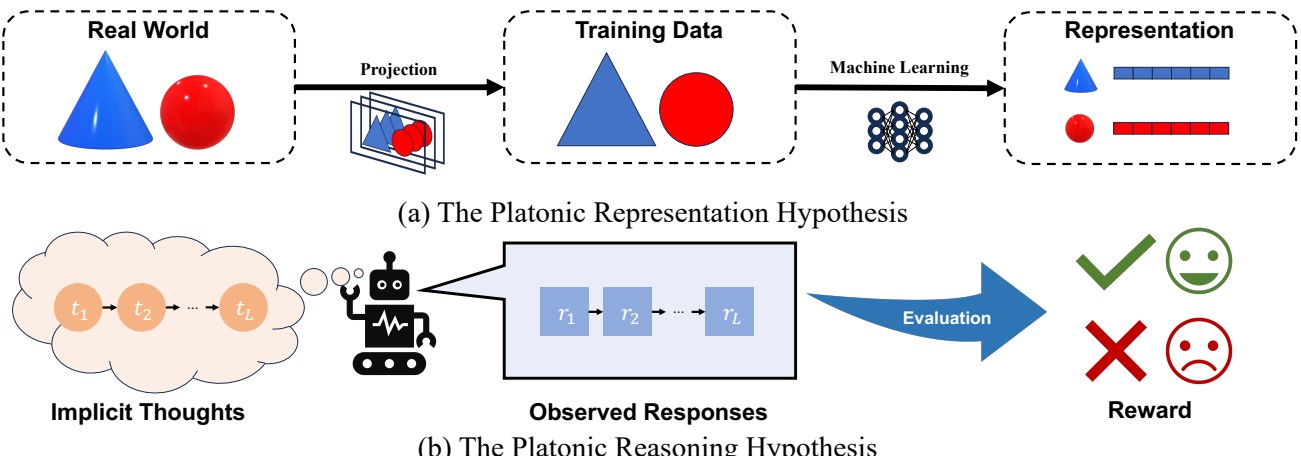

(a) The Platonic Representation Hypothesis

(b) The Platonic Reasoning Hypothesis

*Figure 1.* Two examples of Plato's allegory of the cave in artificial intelligence. (a) The training data accessible to the model are just shadows on the wall, serving as projections of the real world. (b) The responses of the LLM we observe are also shadows on the wall, reflecting the model's implicit reasoning thoughts during task execution.

or ranking (Cobbe et al., 2021). Beyond simpler methods, advanced frameworks like CoT (Wei et al., 2022), ToT (Yao et al., 2024), and MCTS-based approaches inspired by AlphaGo (Silver et al., 2016) explore solution spaces in tree structures to identify optimal answers (Zhang et al., 2024; Feng et al., 2023).

Despite their promise, external slow-thinking methods face several challenges. **First,** the mechanisms behind their effectiveness remain poorly understood, hindering the design of more advanced and efficient strategies. **Second,** practical implementations of complex slow-thinking techniques often achieve limited success unless significant computational resources are added. This is due to the difficulty of optimizing design choices and hyperparameters, which frequently results in suboptimal performance. To address these challenges, we propose a systematic framework based on information theory, linking external slow-thinking methods to the probability of correct reasoning in LLMs.

We begin by analyzing the snowball errors in LLM reasoning in Section 2, and subsequently relate this effect to the likelihood of reasoning errors in Section 3. In Section 4, we further explore the probability of correct reasoning within the the real-world practice, and compare various slow-thinking strategies in Section 5. Finally, we review relevant literature in Section 6 and present our conclusions in Section 7.

## 2. Snowball Errors in LLM Reasoning

Imagine rolling a snowball on a snowy surface during winter. As the distance increases, the snowball grows at an accel-

erating rate. This "snowball effect" illustrates how small changes can compound over time. In the context of LLMs, this effect first manifests as the progressive accumulation of token-level errors in auto-regressive next-token prediction (NTP) tasks, potentially causing significant deviations from the expected or golden answers (Bachmann & Nagarajan, 2024).

For reasoning tasks, however, the snowball effect shifts to the sentence level, making the errors more challenging to characterize. To understand these errors, it is critical to first examine the nature of reasoning. Prior research suggests that LLM reasoning can be conceptualized as executing a sequence of primitive tasks at each reasoning step (Ton et al., 2024), prompting further investigation into how such errors accumulate across the reasoning process.

Let's reconsider Plato's allegory of the cave[1], which has been widely used to highlight the limitations of AI models (Huh et al., 2024). In this analogy, training data serve as mere projections of the real world, akin to shadows on the wall, as illustrated in Figure 1(a). Similarly, in LLM reasoning, generated responses are the shadows, reflecting the model's implicit reasoning processes, as illustrated in Figure 1(b).

For example, when solving a problem like "Calculate $3x + 2y$," the model implicitly executes reasoning steps such as $t_1$: {Calculate $3x$} $\rightarrow t_2$: {Calculate $2y$} $\rightarrow t_3$: {Add $3x$ and $2y$}. However, these steps are abstract and cannot be directly observed in outputs. Instead, the response sequence $r_1 \rightarrow$

---

[1]In this allegory, Plato describes individuals confined to a cave, learning about the world solely through the shadows on its wall.

$r_2 \rightarrow r_3$ can be multiple possible expressions of the same reasoning process. **Moreover, since individual responses $r_l$ cannot fully encapsulate the corresponding steps $t_l$, minor inaccuracies accumulate, ultimately leading to significant snowball errors.**

To quantify snowball errors in LLM reasoning, we consider mutual information (MI) between the implicit reasoning sequence $t$ and the observed response sequence $r$, denoted as $I(t; r)$. This metric captures the shared information between the two sequences. Furthermore, the minor inaccuracies in the responses at each reasoning step can be assessed as information loss, which can be quantified by the difference between the MI $I(t; r)$ and the information entropy of the implicit thoughts $t$, denoted as $H(t)$. And it can be mathematically defined as:

**Definition 2.1.** (Information loss.) Given a reasoning process with implicit thoughts $t$ and corresponding responses $r$, the information loss in the $l$-th step is defined as:

$$\text{InfoLoss}(r_l) = H(t_l) - I(t_l; r_l) = H(t_l|r_l).$$

The snowball errors can be further defined as the accumulation of information loss across all reasoning steps as follows:

**Definition 2.2.** (Snowball errors, or cumulative information loss.) Given a reasoning process with implicit thoughts $t$ and corresponding responses $r$, the snowball errors in the $l$-th step are defined as:

$$H_{<l}(t|r) = \sum_{i}^{l-1} H(t_i|r_i),$$

where $l$ denotes the number of reasoning steps.

# 3. From Snowball Errors to Probability

In this section, we seek to establish a theoretical connection between snowball errors and the probability of reasoning errors in LLMs. We begin by formally defining the probability of reasoning errors and subsequently derive a lower bound for this probability using principles from information theory. Finally, we empirically validate the presence of snowball errors in the reasoning processes of LLMs.

## 3.1. Probability of Reasoning Errors

As the reasoning path grows longer, snowball errors accumulate, leading to significant factual inaccuracies, which we define as reasoning errors. This subsection explores the relationship between snowball errors and the probability of reasoning errors.

To evaluate reasoning errors, we first define them clearly. Since the response $r_l$ represents the implicit thought $t_l$, a natural approach is to assess whether a sufficiently powerful mapping function $f$ can reconstruct $t_l$ from $r_l$.

**Proposition 3.1.** *(Probability of reasoning errors.) Let $t_l$ denote an implicit thought at step $l$, and $r_l$ represent the corresponding generated response. Given a predicted thought $\hat{t}_l$ derived from $r_l$ using a prediction function $\hat{t}_l = f(r_l)$, the probability of reasoning error at step $l$ is defined as the likelihood of the event $e_l$, where $e_l : \hat{t}_l \neq t_l$. This probability is denoted as $P(e_l) = P(\hat{t}_l \neq t_l)$.*

To estimate the probability of reasoning errors, we propose utilizing information theory to establish a connection between snowball errors and the likelihood of reasoning errors. Specifically, our analysis start from the following lemma:

**Lemma 3.2.** *(Information loss inequality.) Given a reasoning process defined above, and under the assumption that the mutual information $I(t_l; r_l)$ decreases with respect to $l$ when $l \geq 2$, the information loss in the $l$-th step satisfies:*

$$H(t_l|r_l) \geq \frac{H_{<l}(t|r)}{l - 1}.$$

The proof is provided in Appendix A.1. Lemma 3.2 indicates that the information loss in the $l$-th step is bounded by the average snowball errors in the previous steps. Based on this lemma, we can subsequently derive the lower bound of the probability of reasoning errors.

**Theorem 3.3.** *(Lower bound of $P(e_l)$.) Given a reasoning process and conditions defined above, when $l \geq 2$, the probability of reasoning error at step $l$ satisfies:*

$$P(e_l) \geq \log^{-1}(|\mathcal{T}_l| - 1) \left[ \frac{H_{<l}(t|r)}{l - 1} - H_b(e_l) \right],$$

*where $|\mathcal{T}_l|$ is the size of the support of $t_l$. $H_b(e_l)$ is the entropy of the indicator random variable of event $e_l$, which is a relatively small constant.*

The proof mainly relies on Fano's inequality (Fano, 2008) and is provided in Appendix A.2. Theorem 3.3 sets a lower bound on the probability of reasoning errors at every intermediate step $l$. Since $H_{<l}(t|r)$ represents the cumulative information loss up to step $l$, it is assumed to increase at least linearly with $l$. **Furthermore, when the snowball effect occurs, $H_{<l}(t|r)$ may grow faster than linearly.** As a result, the lower bound on the probability of reasoning errors rises with the reasoning path length $l$, indicating that the chance of errors increases as snowball errors accumulate.

## 3.2. Empirical Verification

To empirically verify the existence of snowball errors in LLM reasoning, we conducted experiments using the GSM8k dataset (Cobbe et al., 2021). The evaluation was performed on three state-of-the-art reasoning LLMs: Llama-3.1-8B-Instruct (2024), Qwen2.5-7B-Instruct (2024a), and Skywork-o1-Open-Llama-3.1-8B (2024). We estimate the

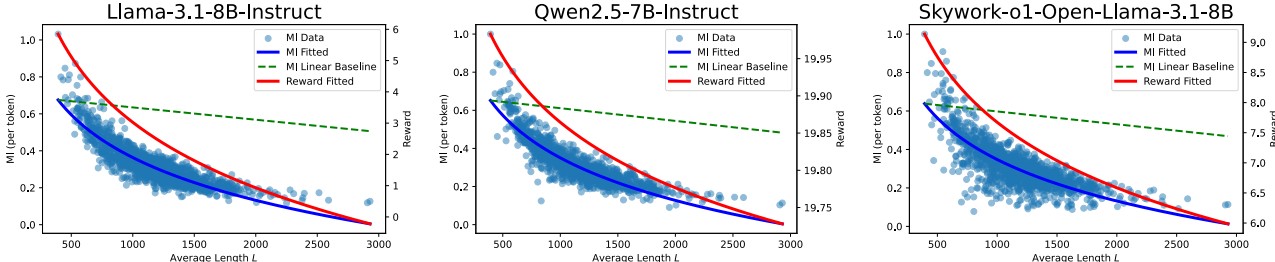

*Figure 2.* The estimated MI and reward of the responses generated by different LLMs on GSM8k. The x-axis denotes the average length of the responses. The left y-axis represents the estimated MI between each response and the golden answer, whereas the right y-axis shows the corresponding reward. "•" indicates the estimated MI for individual questions, while "——" and "——" respectively depict the fitted curves of the estimated MI and reward. "- - -" depicts the linear baseline.

mutual information $I(t; r)$ per token and the reward of the responses to illustrate their relationship with the reasoning path length $L$. Detailed experimental settings are provided in Appendix C.1.

As illustrated in Figure 2, the estimated mutual information decreases in a nearly exponential and much faster rate than linear decay as $L$ increases. Since our verification computes the average mutual information per token, the actual rate of decline in mutual information for tokens appearing later in the sequence is likely to be more pronounced. Furthermore, the reward scores of the responses correspond to the mutual information, and also diminish with increasing response length. These findings confirm the existence of snowball errors by verifying that the information loss can accumulate much faster than linearly in the reasoning processes of LLMs, and illustrate their relevance to the quality of responses, aligning with our theoretical analysis. For a more robust verification, we also include results on larger LLMs for a more robust verification in Appendix D.1 and an additional analysis by different difficulty levels in Appendix D.2.

## 4. Probability of Correct Reasoning in External Slow-Thinking

Previous analyses demonstrate that the probability of reasoning errors, $P(e_l)$, increases with the number of reasoning steps $l$. In the practice, however, reasoning errors are often reflected in the reward associated with the generated responses. In this section, we extend our theoretical analysis to real-world scenarios and explore the mechanisms underlying the effectiveness of external slow-thinking methods.

### 4.1. What is a Correct Reasoning?

We begin our analysis by defining the reasoning process in real-world settings. Given a question $r_0$, a response $\mathcal{R}$ is

represented as a sequence of $L$ reasoning steps, i.e., $\mathcal{R} = [r_1, r_2, \cdots, r_L]$, generated autoregressively by the LLM. An oracle $\phi$ is then employed to evaluate the quality of each step $r_l$ produced at layer $l$, denoted as $\phi(r_l)$. In practice, this evaluation is often determined by human feedback or a reward model. Furthermore, we assume that for each reasoning step, there exists a corresponding golden step $r_l^*$, representing the most accurate and correct step that the LLM should generate, aligning with ideal human reasoning.

Based on the above setting, the oracle evaluation can be used to quantify the correctness of a response. Specifically, we define this measure as follows:

**Definition 4.1.** ($\tau$-correct step.) A step $r_l$ is considered as $\tau$-correct if the quality difference between the step and the golden step is less than $\tau$, i.e., $|\phi(r_l) - \phi(r_l^*)| \leq \tau$.

Similarly, we can define the correctness of an entire reasoning process as follows:

**Definition 4.2.** ($\tau$-correct reasoning.) A response $\mathcal{R}$ is considered as $\tau$-correct if all steps in the sequence are $\tau$-correct, i.e., $\forall l, |\phi(r_l) - \phi(r_l^*)| \leq \tau$ (denoted as $\psi(\mathcal{R}) \leq \tau$).

Definition 4.1 and Definition 4.2 provide formal definitions for measuring the correctness of a reasoning step or an entire reasoning process. Intuitively, for a given task, the probability of achieving a $\tau$-correct step is determined by three primary factors: the capability of the LLM, the correctness threshold $\tau$, and the length of the current reasoning path.

### 4.2. Probability of Correct Reasoning

The results shown in Figure 2 indicate that the average MI can decrease exponentially with the reasoning length, which suggests that the average snowball errors increase exponentially with the reasoning length. Since the probability cannot exceed 1, with Theorem 3.3, we thus hypothesize that the probability of encountering an error in practice follows an

exponential decay function: $P(e_l) = 1 - \lambda e^{-l}$ for the convenience of subsequent analyses. Thus the probability of generating a correct step in layer $l$ can be proposed as:

**Proposition 4.3.** *(Probability of $\tau$-correct step.) We propose that the probability of generating a $\tau$-correct step is related to the layer index $l$ in the following way:*

$$\Pr\left[|\phi(r_l) - \phi(r_l^*)| \leq \tau\right] = \min(\lambda_\tau e^{-l}, 1),$$

*where $\lambda_\tau$ is a constant relevant to correctness $\tau$, and a premise is that the step $r_{l-1}$ is already $\tau$-correct.*

This proposition mainly aims to obtain more intuitive conclusions in subsequent analysis and is based on the intuition of LLM's multi-step reasoning errors. In particular, we point out in Appendix B that under a more relaxed and generalized proposition, our subsequent theoretical results can still be guaranteed.

With Proposition 3.1, we can derive the probability of generating a $\tau$-correct response $\mathcal{R}$ as follows:

**Lemma 4.4.** *(Probability of $\tau$-correct reasoning.) The probability of generating a $\tau$-correct response $\mathcal{R}$ is:*

$$\Pr\left[\psi(\mathcal{R}) \leq \tau\right] = \prod_{l=1}^{L} \Pr\left[|\phi(r_l) - \phi(r_l^*)| \leq \tau\right]$$
$$\leq \lambda_\tau^L e^{-\frac{L(L+1)}{2}}.$$

Lemma 4.4 shows that the probability of generating a correct response decreases exponentially with the reasoning leangth $L$. This result aligns with the practical experience that the LLM is more likely to make mistakes in more complex reasoning tasks with more reasoning layers.

### 4.3. External Slow-Thinking Mechanisms

In general, external slow-thinking methods aim to enhance the correctness of generated responses by incorporating additional reasoning steps. Due to the inherent stochastic nature of the LLM's sampling mechanism, the probability of generating a correct response cannot be guaranteed. However, by introducing supplementary reasoning steps and employing multiple re-sampling strategies, the likelihood of producing a correct response can be effectively increased.

Although different methods employ varying strategies to explore the reasoning space, they share two common characteristics: (1) *Width-Expansion.* For a reasoning sequence of length $L$, most external slow-thinking methods aim to expand the width of the reasoning space. This expansion is achieved either through simple re-generation techniques (e.g., BoN, CoT-SC) or via more sophisticated approaches like tree search (e.g., ToT, MCTS). (2) *Generation & Selection.* Despite the challenge of generating a good reasoning

step, the expansion of the reasoning space introduces the challenge of selecting the most promising reasoning path from a pool of candidates. In summary, let $\Pr(\tau_{\text{generate}})$ denote the probability of generating a $\tau$-correct reasoning and $\Pr(\tau_{\text{select}})$ denote the probability of selecting a $\tau$-correct reasoning. The overall probability of obtaining a $\tau$-correct response can then be expressed as:

$$\Pr\left[\psi(\mathcal{R}) \leq \tau\right] = \Pr\left(\tau_{\text{gennerate}}\right) \times \Pr\left(\tau_{\text{select}}\right).$$

While the width-expansion strategy can effectively increase $\Pr\left(\tau_{\text{generate}}\right)$, the additional reasoning steps introduced also heighten the complexity of selecting the most promising reasoning path, thereby reducing $\Pr\left(\tau_{\text{select}}\right)$. To illustrate this trade-off, we use beam search as a baseline width-expansion strategy. Beam search is a widely adopted method in tree search algorithms, where the search tree's width is expanded by generating $k$ child nodes at each layer while retaining only the top-$b$ most promising candidates. The result of this analysis is formalized in the following lemma.

**Lemma 4.5.** *(Probability of $\tau$-correct reasoning in width-expanding methods.) For a beam-search-like strategy which samples $k$ steps and keeps $b$ steps as candidates at each expansion, the probability of obtaining a $\tau$-correct response is upper bounded by:*

$$\Pr\left[\psi(\mathcal{R}) \leq \tau\right] \leq \prod_{l=1}^{L} \epsilon_b[1 - (1 - \lambda_\tau e^{-l})^k],$$

*where $\epsilon_b$ is the probability of selecting a $\tau$-correct step from $b$ candidates, which is primarily determined by the reliability of the value function employed in the selection.*

The proof is provided in Appendix A.3. Since the typical reasoning setting holds that $k \geq 1$, and $0 \leq \lambda_\tau e^{-l} \leq 1$, the probability of generating a correct response can be further simplified as follows:

**Theorem 4.6.** *(Upper bound of the probability of $\tau$-correct reasoning in width-expanding methods.) The probability of obtaining a $\tau$-correct response is upper bounded by:*

$$\Pr\left[\psi(\mathcal{R}) \leq \tau\right] \leq \epsilon_b^L k^L \lambda_\tau^L e^{-\frac{L(L+1)}{2}}.$$

The proof is provided in Appendix A.4, and we also provide an empirical analysis about the influence of $k$ in Appendix D.3. Lemma 4.5 and Theorem 4.6 show that the probability of generating a correct response in width-expanding methods depends on three key factors: the number of generations $k$ per layer, the number of candidates $b$ selected per layer, and the correctness threshold $\tau$. Specifically, components related to $\epsilon_b$ represent $\Pr\left(\tau_{\text{select}}\right)$, which are influenced by the reliability of the value function used in selection. Meanwhile, $\Pr\left(\tau_{\text{generate}}\right)$ is mainly determined by

the expansion width $k$ and reasoning path length $L$. Compared with Lemma 4.4, Theorem 4.6 further illustrates the mechanism of external slow-thinking methods. By scaling $k$, these methods can improve the probability of correct reasoning at the cost of additional reasoning steps. However, the effectiveness heavily depends on the reliability of the value function, reflected in $\epsilon_b^L$. For slow-thinking methods to be effective, the value function must satisfy $\epsilon_b > \frac{1}{k}$ to ensure an improvement in the probability upper bounds.

Based on the above analysis, we can infer the underlying mechanisms of external slow-thinking methods. **By expanding the reasoning space, these methods effectively increase the probability of generating a correct response, thereby mitigating the impact of snowball errors.** However, selecting the most promising reasoning path poses a significant challenge, **as the effectiveness of this selection heavily depends on the reliability of the employed value function, which can substantially influence the overall performance of the method.** We also provide analyses of the mechanism of external slow-thinking methods from other perspective in Appendix E.

# 5. Comparison between External Slow-Thinking Frameworks

Previous analyses have demonstrated that external slow-thinking methods effectively increase the search width, thereby enhancing the probability of correct reasoning. In this section, we use the simplest strategy, Best-of-N (BoN), and the widely adopted sophisticated strategy, Monte Carlo Tree Search (MCTS), as examples to compare their effectiveness and examine the impact of specific framework designs.

## 5.1. Correct Reasoning Probability

We begin by determining the probability of correct reasoning for BoN and MCTS using the results from Theorem 4.6. With the assumption that the total number of reasoning steps is $L$, BoN can be characterized as generating $N$ reasoning steps in the first layer, followed by the generation of a single step in subsequent layers, and finally applying a reward model (RM) to select one path from the $N$ candidates in the $L$-th layer. Accordingly, the probability of achieving $\tau$-correct reasoning with BoN can be bounded as follows:

**Lemma 5.1.** *(Upper bound of the probability of $\tau$-correct reasoning with BoN.) The probability of obtaining a $\tau$-correct response in BoN satisfies:*

$$\Pr\left[\psi(\mathcal{R}) \leq \tau\right] \leq \epsilon_N N^L \lambda_\tau^L e^{-\frac{L(L+1)}{2}}.$$

In contrast, MCTS employs a more intricate structure, making it difficult to derive a closed-form expression for the probability of correct reasoning. To simplify the analysis, we consider the "best-case" and "worst-case" scenarios for

MCTS. Here, the "best" and "worst" cases are defined based on the difficulty for BoN to achieve a comparable probability of correct reasoning, rather than the actual performance of MCTS.

Using the RAP-like classic MCTS strategy (Hao et al., 2023) as an example, where MCTS terminates once a reasoning path of length $L$ is derived, **the best-case scenario occurs when MCTS expands only the deepest leaf node at each step.** Assuming $b$ child nodes are expanded per iteration, MCTS in this scenario reduces to a beam search strategy with both a sample size and beam width of $b$. The probability of obtaining a $\tau$-correct reasoning in the best-case scenario can then be expressed as:

**Lemma 5.2.** *(Upper bound of the probability of $\tau$-correct reasoning with MCTS in best case.) The probability of obtaining a $\tau$-correct response in MCTS in the best case satisfies:*

$$\Pr\left[\psi(\mathcal{R}) \leq \tau\right] \leq \epsilon_b^L b^L \lambda_\tau^L e^{-\frac{L(L+1)}{2}}.$$

**In the worst-case scenario, MCTS must expand all possible child nodes at each step, ultimately constructing a complete $b$-ary search tree** that terminates at the $L$-th layer. Consequently, the probability of obtaining a $\tau$-correct response with MCTS in the worst-case scenario can be bounded as:

**Lemma 5.3.** *(Upper bound of the probability of $\tau$-correct reasoning with MCTS in worst case.) The probability of generating a $\tau$-correct response in MCTS in the worst case satisfies:*

$$\Pr\left[\psi(\mathcal{R}) \leq \tau\right] \leq \lambda_\tau^L \left(\frac{e}{b}\right)^{-\frac{L(L+1)}{2}} \prod_{l=1}^{L} \epsilon_{b^l}.$$

The proof is analogous to that of Theorem 4.6, with the primary difference being that, in Lemma 5.2, we have $k = b$, whereas in Lemma 5.3, $k = b^l$.

Lemma 5.2 and Lemma 5.3 establish upper bounds on the probability of correct reasoning for MCTS in the best-case and worst-case scenarios, respectively. These results illustrate that the probability of correct reasoning with MCTS is influenced by both the search width $b$ and the length of the reasoning path $L$. Although MCTS provides a more sophisticated reasoning strategy and introduces additional complexity to mitigate snowball errors, it is important to note that selection errors also accumulate due to the increased complexity. This accumulation is reflected in the exponential term within the $\epsilon$ terms.

## 5.2. Comparison between BoN and MCTS

Since MCTS introduces exponential selection errors due to its increased complexity, it is necessary to perform a fair

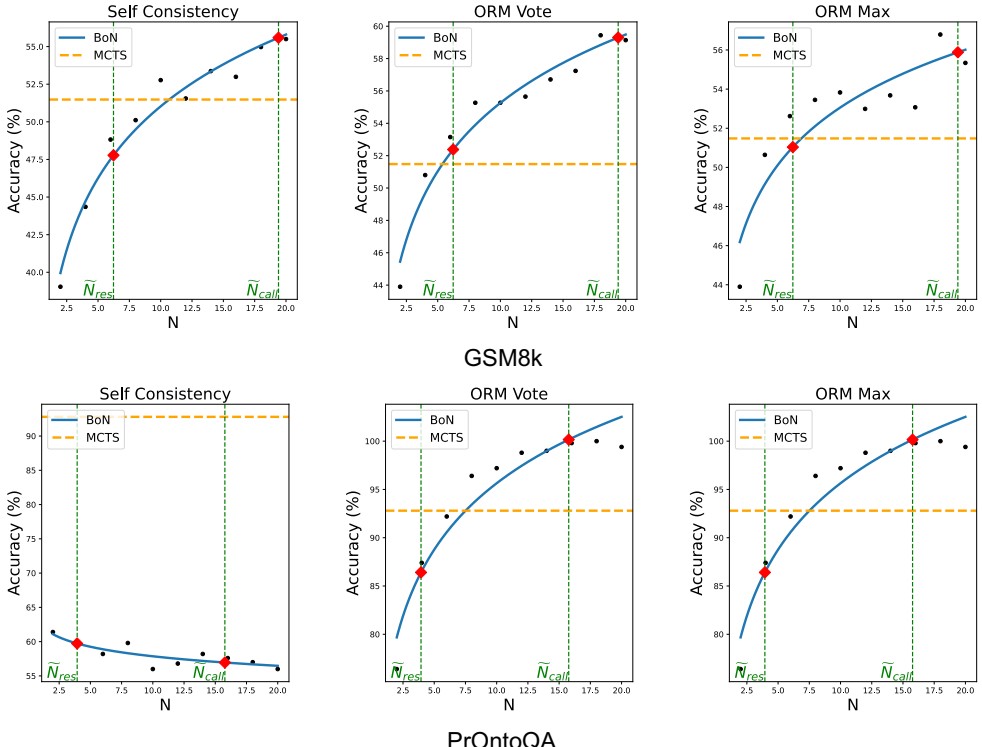

*Figure 3.* Comparison of the accuracy of various reasoning strategies on GSM8k and PrOntoQA. "- - -" indicates the accuracy of the baseline MCTS, while "——" represents the accuracy of BoN under different $N$ settings. The x-axis corresponds to the varying values of $N$. Additionally, the estimated values $\widetilde{N}_{res}$ and $\widetilde{N}_{call}$ are marked by vertical green dashed lines.

comparison between BoN and MCTS. This challenge arises from the inability to fully characterize the reward model (RM) employed in practice. For instance, it is difficult to determine whether selecting a correct step from $N$ candidates in a single iteration is more or less challenging than selecting a correct step from $b$ candidates across $L$ iterations, particularly when $N = O(b^L)$.

To address this issue, we compare the two methods by examining the minimal value of $N$ required for BoN to achieve a probability of correct reasoning comparable to MCTS in both the best-case and worst-case scenarios, assuming the use of an ideal reward model, which ensures that whenever at least one correct candidate exists, $\forall k, \epsilon_k = 1$.

Under this assumption, we derive the minimal $N$ required for BoN to match the probability of correct reasoning with MCTS in the best-case scenario by comparing the upper bounds provided in Lemma 5.1 and Lemma 5.2:

**Corollary 5.4.** *Despite the influence of RM, the minimal $N$ for making BoN obtain the comparable correct reasoning probability with MCTS in best case is $N_{min}^{best} = O(b)$.*

Similarly, we can derive the minimal $N$ required for BoN

to match the probability of correct reasoning with MCTS in the worst-case scenario by comparing the upper bounds provided in Lemma 5.1 and Lemma 5.3:

**Corollary 5.5.** *Despite the influence of RM, the minimal $N$ for making BoN obtain the comparable correct reasoning probability with MCTS in worst case is $N_{min}^{worst} = O(b^{\frac{L}{2}})$.*

Corollary 5.4 and Corollary 5.5 reveal that the minimal $N$ required for BoN to achieve a probability of correct reasoning comparable to MCTS is $O(b)$ in the best-case scenario and $O(b^{\frac{L}{2}})$ in the worst-case scenario.

Next, we analyze the total reasoning cost of BoN and MCTS when $N$ is set with corresponding $N_{min}$. Since BoN generates $N$ reasoning paths, each of length $L$, the total reasoning cost for BoN is $N \times L$. Using the results from the above corollaries, we can infer that when MCTS operates in the best-case scenario, the total reasoning cost of BoN is $O(bL)$. In contrast, when MCTS is in the worst-case scenario, the total reasoning cost of BoN increases to $O(Lb^{\frac{L}{2}})$. For MCTS, according to previous analyses, the total reasoning cost is $O(bL)$ in the best-case scenario and $O(b^L)$ in the worst-case scenario. We summarize the comparison of the total

|            | **BoN** (comparable) | **MCTS** (baseline) |
|------------|:--------------------:|:-------------------:|
| Best Case  | $O(bL)$              | $O(bL)$             |
| Worst Case | $O(Lb^{\frac{L}{2}})$ | $O(b^L)$            |

*Table 1.* Comparison of the total reasoning cost between MCTS and its comparable BoN. We assume MCTS generates a reasoning path of length $L$ and expands $b$ child nodes at each expansion, while BoN generates $N_{\min}$ reasoning paths, each of length $L$.

reasoning cost between BoN and MCTS in Table 1.

The results in Table 1 indicate that when BoN and MCTS achieve a comparable probability of correct reasoning, the total reasoning cost of BoN remains close to that of MCTS. In the best-case scenario, their costs are asymptotically equivalent, while in the worst-case scenario, BoN's cost may exceed MCTS when $L$ is small but stays reasonable. As $L$ grows larger, BoN's cost can even fall below MCTS. This analysis shows that BoN can achieve similar reasoning accuracy to MCTS with comparable reasoning costs, and these findings are applicable to other external slow-thinking frameworks.

In conclusion, **external slow-thinking methods introduce additional reasoning steps to mitigate the impact of snowball errors.** However, on one hand, the **inaccuracy of the reward function can result in the additional reasoning steps incurring extra selection costs**, which may decrease the probability of correct reasoning. On the other hand, **the effectiveness of mitigating snowball errors is primarily determined by the total reasoning cost**, with the specific framework having limited impact on the overall outcome.

### 5.3. Empirical Evaluation

We empirically compare the accuracy of BoN and MCTS on two reasoning tasks: GSM8k (Cobbe et al., 2021) and PrOntoQA (Saparov & He, 2022). Following prior work (Feng et al., 2023), we use the recommended settings to optimize MCTS and calculate the corresponding $N$ for BoN to align its reasoning cost with MCTS. Due to differences in how reasoning paths are generated, exact alignment is impractical. To address this, we define $N$ as reasonable if it lies between $\widetilde{N}_{\text{res}}$ (aligned reasoning steps) and $\widetilde{N}_{\text{call}}$ (aligned LLM calls). We evaluate BoN with three selection strategies: Self-Consistency, ORM Vote, and ORM Max. Details of the experimental setup are in Appendix C.2, and results are presented in Figure 3. Furthermore, we also include additional experimental verifications on a planning task Game24 (Yao et al., 2024) in Appendix D.4.

Since PrOntoQA is a binary classification task with only true or false answers, increasing $N$ in the Self-Consistency strategy cannot improve BoN's performance without a reward model. In contrast, for GSM8k, where answers are diverse,

increasing $N$ can enhance BoN's performance even without a reward model. For ORM Vote and ORM Max strategies, guided by the reward model, BoN achieves performance comparable to MCTS when $N$ lies between $\widetilde{N}_{\text{res}}$ and $\widetilde{N}_{\text{call}}$. Notably, when $N$ is near $\widetilde{N}_{\text{res}}$, BoN may slightly underperform compared to MCTS but not significantly. Conversely, setting $N$ to a larger value within this range allows BoN to match or even surpass MCTS. These findings align with previous observations of MCTS's limited success in LLM reasoning and support our theoretical analysis.

## 6. Related Work

**Information Theory.** Information theory provides a theoretical basis for quantifying the information contained in random variables. The entropy $H(X)$ of a random variable $X$ measures its information content, while mutual information $I(X;Y) = H(X) - H(X|Y)$ quantifies the information shared between two variables. Applications of information theory span numerous fields, including bounding the generalization capacity of deep learning models (Russo & Zou, 2019; Xu & Raginsky, 2017) and improving task interpretability (Slonim et al., 2001; Hu et al., 2019; West et al., 2019). Recently, it has been employed to analyze synthetic data generation (Gan & Liu, 2024) and measure reasoning errors in LLMs (Ton et al., 2024).

**Reasoning with LLMs.** LLMs have made substantial progress in understanding and generation, particularly for complex reasoning tasks. Foundational work (Brown et al., 2020) established LLMs as few-shot learners, while Chain-of-Thought (CoT) prompting (Wei et al., 2022) introduced multi-step reasoning by explicitly generating intermediate steps. Self-Consistency (Wang et al., 2022) further improved robustness by aggregating multiple reasoning paths. Information-theoretic approaches (Ton et al., 2024) have also provided theoretical insights into quantifying and mitigating reasoning errors, showcasing the interplay between empirical advancements and theoretical frameworks.

**External Slow-Thinking.** Test-time scaling has proven effective in enhancing LLM reasoning (Snell et al., 2024), this timely inference scaling law has been empirically analyzed and verified (Wu et al., 2024). External slow-thinking methods, such as generating additional tokens (Wei et al., 2022) and incorporating tree search algorithms (Yao et al., 2024), expand the reasoning space without retraining. Techniques like beam search (Kang et al., 2024) and Monte Carlo Tree Search (MCTS) (Zhang et al., 2024; Feng et al., 2023) have further advanced reasoning capabilities. However, these methods require significantly more computational resources compared to simpler strategies like Best-of-N and often yield limited practical improvement. The underlying mechanisms of their effectiveness remain an area of ongoing exploration.

# 7. Conclusion

In this paper, we analyzed the mechanisms behind the effectiveness of external slow-thinking methods. We linked snowball errors in LLM reasoning to reasoning errors using information theory and showed how external slow-thinking methods reduce errors by expanding the reasoning space. We also examined the trade-off between additional reasoning costs and the probability of achieving correct reasoning. Through comparisons of methods ranging from Best-of-N (BoN) to Monte Carlo Tree Search (MCTS), we found that the key factors influencing effectiveness are the reward function's capability and the total reasoning cost, with the specific search framework playing a secondary role. Our findings suggest that optimizing reward functions and improving policy model reasoning capabilities are more essential for designing more effective external slow-thinking methods in a long run.

# Acknowledgments

This research was supported by National Natural Science Foundation of China (No.62476277), National Key Research and Development Program of China (No. 2024YFE0203200), CCF-ALIMAMA TECH Kangaroo Fund (No.CCF-ALIMAMA OF 2024008), and Huawei-Renmin University joint program on Information Retrieval. We also acknowledge the support provided by the fund for building worldclass universities (disciplines) of Renmin University of China and by the funds from Beijing Key Laboratory of Big Data Management and Analysis Methods, Gaoling School of Artificial Intelligence, Renmin University of China, from Engineering Research Center of Next-Generation Intelligent Search and Recommendation, Ministry of Education, from Intelligent Social Governance Interdisciplinary Platform, Major Innovation & Planning Interdisciplinary Platform for the "DoubleFirst Class" Initiative, Renmin University of China, from Public Policy and Decision-making Research Lab of Renmin University of China, and from Public Computing Cloud, Renmin University of China.

# Impact Statement

This paper presents work whose goal is to advance the field of Large Language Models. There are many potential societal consequences of our work, none which we feel must be specifically highlighted here.

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

# A. Proofs

## A.1. Proof of Lemma 3.2

*Proof.* Since $I(t_l; r_l)$ decreases with respect to $l$, we have

$$I(t_l; r_l) \leq \frac{\sum_i^{l-1} I(t_i; r_i)}{l-1},$$

by the definition of mutual information, we can derive that

$$H(t_l) - H(t_l|r_l) \leq \sum_i^{l-1} \left[ \frac{H(t_i) - H(t_i|r_i)}{l-1} \right]$$

$$H(t_l|r_l) \geq \sum_i^{l-1} \frac{H(t_i|r_i)}{l-1} + H(t_l) - \sum_i^{l-1} \frac{H(t_i)}{l-1} \tag{1}$$

Since along the reasoning process, LLM is continiously introducing new information, we have $H(t_l) \geq \sum_i^{l-1} \frac{H(t_i)}{l-1}$. Therefore, we have:

$$\sum_i^{l-1} \frac{H(t_i|r_i)}{l-1} + H(t_l) - \sum_i^{l-1} \frac{H(t_i)}{l-1} \geq \sum_i^{l-1} \frac{H(t_i|r_i)}{l-1}. \tag{2}$$

Hence, together with equation (1) and equation (2), we can further derive that:

$$H(t_l|r_l) \geq \sum_i^{l-1} \frac{H(t_i|r_i)}{l-1} := \frac{H_{<l}(t|r)}{l-1}. \tag{3}$$

This finishes the proof. $\square$

## A.2. Proof of Theorem 3.3

*Proof.* Define an indicator random variable $E$, where $E = 1$ if $e_l$ occurs, and $E = 0$ otherwise. More specifically,

$$E := \begin{cases} 1 & \text{if } \hat{t}_l \neq t_l, \\ 0 & \text{if } \hat{t}_l = t_l. \end{cases}$$

Based on the definition of entropy, we have:

$$H(t_l|\hat{t}_l) = H(E|\hat{t}_l) + H(t_l|E, \hat{t}_l). \tag{4}$$

For the first term $H(E|\hat{t}_l)$, we have:

$$H(E|\hat{t}_l) \leq H(E) := H_b(e_l). \tag{5}$$

For the second term $H(t_l|E, \hat{t}_l)$, we can expand it as:

$$H(t_l|E, \hat{t}_l) = H(t_l|E = 0, \hat{t}_l)P(E = 0) + H(t_l|E = 1, \hat{t}_l)P(E = 1). \tag{6}$$

Since $E = 0$ implies $\hat{t}_l = t_l$, we have $H(t_l|E = 0, \hat{t}_l) = 0$. And since $E = 1$ implies $\hat{t}_l \neq t_l$, we have $P(E = 1) = P(e_l)$. Therefore, we have:

$$H(t_l|E, \hat{t}_l) = H(t_l|E = 1, \hat{t}_l)P(e_l). \tag{7}$$

Considering the upper bound of the entropy of a random variable, we have the following lemma.

**Lemma A.1.** *(Principle of maximum entropy.) Let $X$ be a random variable with support $\mathcal{X}$, the entropy $H(X)$ satisfies:*

$$H(X) \leq \log(|\mathcal{X}|),$$

*with equality if and only if $X$ is uniformly distributed over $\mathcal{X}$.*

Since $E = 1$ indicates that $\hat{t}_l \neq t_l$, then when $\hat{t}_l$ is given as condition, random variable $t_l$ can be narrowed down to $|\mathcal{T}_l| - 1$ different values, where $\mathcal{T}_l$ is the support of $t_l$. Therefore, we have:

$$H(t_l|E = 1, \hat{t}_l) \leq \log(|\mathcal{T}_l| - 1). \tag{8}$$

Combining equation (7) and equation (8), we have:

$$H(t_l|E, \hat{t}_l) \leq \log(|\mathcal{T}_l| - 1)P(e_l). \tag{9}$$

Combining equation (5) and equation (9), we have:

$$H(t_l|\hat{t}_l) \leq \log(|\mathcal{T}_l| - 1)P(e_l) + H_b(e_l). \tag{10}$$

Considering the markov chain $t_l \rightarrow r_l \rightarrow \hat{t}_l$, due to data processing inequality, we have:

$$I(t_l; \hat{t}_l) \leq I(t_l; r_l).$$

Based on the definition of mutual information, we can further derive that:

$$H(t_l|r_l) \leq H(t_l|\hat{t}_l). \tag{11}$$

With equation (10) and equation (11), we have:

$$H(t_l|r_l) \leq \log(|\mathcal{T}_l| - 1)P(e_l) + H_b(e_l). \tag{12}$$

With lemma 3.2, we have:

$$H(t_l|r_l) \geq \frac{H_{<l}(t|r)}{l-1}. \tag{13}$$

Together with equation (12) and equation (13), we have:

$$\log(|\mathcal{T}_l| - 1)P(e_l) + H_b(e_l) \geq \frac{H_{<l}(t|r)}{l-1}. \tag{14}$$

Hence, we can further derive that:

$$P(e_l) \geq \log^{-1}(|\mathcal{T}_l| - 1)\left[\frac{H_{<l}(t|r)}{l-1} - H_b(e_l)\right]. \tag{15}$$

This finishes the proof. $\qquad\square$

### A.3. Proof of Lemma 4.5

*Proof.* Assuming that the probability of generating a $\tau$-correct thought is $p$, then with $k$ sampling times, the probability of generating at least one $\tau$-correct thought is $1 - (1-p)^k$.

With lemma 4.3, we have $p \leq \lambda e^{-l}$, then the probability of generating a $\tau$-correct thought at the $l$-th layer is

$$\Pr\left[|\phi(r_l) - \phi(r_l^*)| \leq \tau\right] \leq 1 - (1 - \lambda_\tau e^{-l})^k. \tag{16}$$

The probability of generating a $\tau$-correct response is the product of the probability of generating a $\tau$-correct thought in each layer,

$$\Pr\left[\psi(\mathcal{R}) \leq \tau\right] = \prod_{l=1}^{L} \epsilon_b \Pr\left[|\phi(r_l) - \phi(r_l^*)| \leq \tau\right] \leq \prod_{l=1}^{L} \epsilon_b[1 - (1 - \lambda_\tau e^{-l})^k]. \tag{17}$$

$\qquad\square$

### A.4. Proof of Theorem 4.6

*Proof.* The following inequality always holds when $k \geq 1$ and $0 \leq \lambda e^{-l} \leq 1$:

$$(1 - \lambda_\tau e^{-l})^k \geq 1 - k\lambda_\tau e^{-l}. \tag{18}$$

Then with lemma 4.5, we have:

$$
\begin{aligned}
\Pr\left[\psi(\mathcal{R}) \leq \tau\right] &\leq \prod_{l=1}^{L} \epsilon_b [1 - (1 - \lambda_\tau e^{-l})^k] \\
&\leq \prod_{l=1}^{L} \epsilon_b [1 - (1 - k\lambda_\tau e^{-l})] \\
&= \prod_{l=1}^{L} \epsilon_b k \lambda_\tau e^{-l} \\
&= \epsilon_b{}^L k^L \lambda_\tau^L e^{-\frac{L(L+1)}{2}}.
\end{aligned} \tag{19}
$$

This finishes the proof. $\qquad\square$

## B. Relaxed Theoretical Results

Proposition 4.3 was designed to facilitate subsequent analyses while maintaining readability. The negative exponential form provides a simple yet effective characterization of accuracy decay. Below, we would like to prove that our main results remain valid under a weaker assumption when $\Pr\left[|\phi(r_l) - \phi(r_l^*)| \leq \tau\right]$ decreases monotonically with $l$.

**Proposition B.1.** *(Relaxed probability of $\tau$-correct step for Proposition 4.3.)* *Instead of requiring* $\Pr\left[|\phi(r_l) - \phi(r_l^*)| \leq \tau\right] = \min\left(\lambda_\tau e^{-l}, 1\right)$, *we now assume only that the left-hand side decreases monotonically with $l$ and converges to $0$, i.e.,*

$$\Pr\left[|\phi(r_l) - \phi(r_l^*)| \leq \tau\right] = \min\left(\xi(l, \tau), 1\right),$$

*where $\xi(l, \tau) \geq 0$ decreases monotonically with $l$ and converges to $0$.*

Under this weaker condition, by noting that $\xi^L(l, \tau) := \prod_{l=1}^{L} \xi(l, \tau)$, we derive revised relaxed bounds for subsequent results in the main text presented below:

**Lemma B.2.** *(Relaxed probability of $\tau$-correct reasoning for Lemma 4.4.) After relaxation, the probability of generating a $\tau$-correct response $\mathcal{R}$ is:*

$$
\begin{aligned}
\Pr\left[\psi(\mathcal{R}) \leq \tau\right] &= \prod_{l=1}^{L} \Pr\left[|\phi(r_l) - \phi(r_l^*)| \leq \tau\right] \\
&\leq \xi^L(l, \tau).
\end{aligned}
$$

**Lemma B.3.** *(Relaxed probability of $\tau$-correct reasoning in width-expanding methods for Lemma 4.5.) After relaxation, for a beam-search-like strategy which samples $k$ steps and keeps $b$ steps as candidates at each expansion, the probability of obtaining a $\tau$-correct response is upper bounded by:*

$$\Pr\left[\psi(\mathcal{R}) \leq \tau\right] \leq \prod_{l=1}^{L} \epsilon_b \left[1 - (1 - \xi(l, \tau))^k\right].$$

**Theorem B.4.** *(Relaxed upper bound of the probability of $\tau$-correct reasoning in width-expanding methods for Theorem 4.6.) After relaxation, the probability of obtaining a $\tau$-correct response is upper bounded by:*

$$\Pr\left[\psi(\mathcal{R}) \leq \tau\right] \leq \epsilon_b^L k^L \xi^L(l, \tau).$$

**Lemma B.5.** *(Relaxed upper bound of the probability of $\tau$-correct reasoning with BoN for Lemma 5.1.) After relaxation, the probability of obtaining a $\tau$-correct response in BoN satisfies:*

$$\Pr\left[\psi(\mathcal{R}) \leq \tau\right] \leq \epsilon_N N^L \xi^L(l, \tau).$$

**Lemma B.6.** *(Relaxed upper bound of the probability of $\tau$-correct reasoning with MCTS in best case for Lemma 5.2.) After relaxation, the probability of obtaining a $\tau$-correct response in MCTS in the best case satisfies:*

$$\Pr\left[\psi(\mathcal{R}) \leq \tau\right] \leq \epsilon_b^L b^L \xi^L(l, \tau).$$

**Lemma B.7.** *(Relaxed upper bound of the probability of $\tau$-correct reasoning with MCTS in worst case for Lemma 5.3.) After relaxation, the probability of generating a $\tau$-correct response in MCTS in the worst case satisfies:*

$$\Pr\left[\psi(\mathcal{R}) \leq \tau\right] \leq b^{\frac{L(L+1)}{2}} \xi^L(l, \tau) \prod_{l=1}^{L} \epsilon_{b^l}.$$

These modifications of relaxed results preserve the validity of Corollary 5.4, Corollary 5.5, and Table 1, confirming that our theoretical results are robust even without the original exponential form in Proposition 3.1.

## C. Experiment Settings

### C.1. Settings for Verifying Snowball Errors

For each question, we generated multiple responses from the LLMs and collected them as a set to represent $r$. Additionally, a larger LLM, Llama-3.1-Nemotron-70B-Instruct-HF (2024), was used to rewrite the ground-truth answers multiple times, creating a set of rewritten golden answers as an estimate of $t$. We also utilize an outcome reward model Skywork-Reward-Llama-3.1-8B (Liu et al., 2024) to evaluate the quality of the responses. Following prior studies (Qian et al., 2024; Ma et al., 2020), we utilized the Hilbert-Schmidt Independence Criterion (HSIC) (Gretton et al., 2005) as an estimator of mutual information, $I(t_l; r_l)$.

In our experiments, we generated responses for each question in the GSM8k dataset (Cobbe et al., 2021) 10 times using the tested LLMs to construct the response set $r$. Similarly, the ground-truth answers were rewritten 10 times using the Llama-3.1-Nemotron-70B-Instruct-HF model (Wang et al., 2024) to form a set of rewritten golden answers, serving as an approximation of $t$.

The Hilbert-Schmidt Independence Criterion (HSIC) (Gretton et al., 2005) was employed to estimate the mutual information $I(t_l; r_l)$ between the response set and the rewritten golden answers. The bandwidth parameter $\sigma$ for the Gaussian kernel used in HSIC computation was set to 50. To account for differences in response lengths, the estimated HSIC values were further normalized to obtain a per-token measure, ensuring fair comparison across responses of varying lengths.

To ensure consistency in response generation, we utilized the following reasoning prompt for the tested LLMs:

> Please answer the question step by step and put the final answer in \boxed{}.

For rewriting the ground-truth answers, the following rewriting prompt was used:

> You will be given a problem-solving process. Please rewrite this process without changing its logic or content. Ensure that the output includes only the rewritten process and nothing else.
> **Problem-Solving Process:** {input}
> **Rewritten Process:**

This setup was carefully designed to capture the relationship between the generated responses and the golden answers while controlling for logical consistency and content fidelity during the rewriting process.

### C.2. Settings for Comparison between BoN and MCTS

Given a baseline MCTS which expands for $p$ times, and expands $b$ child nodes at each expansion, with the aim to equal the times of calling LLM for inference, we can calculate the corresponding value of $N$ for BoN as $\widetilde{N}_{\text{call}} = p \times b$. Similarly, with the aim to equal the times of reasoning steps, given the average length of reasoning paths is $L$, we can calculate the corresponding value of $N$ for BoN as $\widetilde{N}_{\text{res}} = \frac{p \times b}{L}$.

To ensure a fair and comprehensive comparison, we consider any $N$ within the range of $\widetilde{N}_{call}$ to $\widetilde{N}_{res}$ as a reasonable value for BoN to achieve a comparable reasoning cost to MCTS. Subsequently, we evaluated the performance of BoN using three different selection strategies: (1) Self-Consistency: Select the most frequent final answer among the $N$ reasoning results. (2) ORM Vote: Select the final answer with the highest total ORM scores across all reasoning paths. (3) ORM Max: Select the result of the reasoning path with the highest individual ORM score.

In our experiments, we compared the performance of BoN and MCTS in the context of the Snowball task. We first determine the baseline MCTS setting according to the recommendation in previous work (Feng et al., 2023). Specifically, in GSM8k, we set the tree max width to 6 Tree Max depth to 8. In PrOntoQA, we set the tree max width to 6 and Tree Max depth to 15. And in Game24, we set the tree max width to 20 and Tree Max depth to 4.

Subsequently, we trace the search process of the baseline MCTS and statics the average expansion width $b$ and average expansion times $p$ for the two tasks. We then estimate the ideal average reasoning steps $L$ for the two tasks by analyzing the ground-truth reasonings. Finally, we calculate corresponding $\widetilde{N}_{call}$ and $\widetilde{N}_{res}$ according to above values. The detailed results are shown in Table 2.

|  | GSM8k | PrOntoQA | Game24 |
|---|---|---|---|
| **avg.** $b$ | 4.26 | 1.67 | 4.56 |
| **avg.** $p$ | 4.54 | 9.45 | 3.99 |
| **avg.** $L$ | 3.11 | 4.00 | 3.00 |
| $\widetilde{N}_{\textbf{call}}$ | 19.40 | 15.77 | 18.24 |
| $\widetilde{N}_{\textbf{res}}$ | 6.23 | 3.94 | 6.08 |

*Table 2.* Settings for BoN and MCTS in GSM8k, PrOntoQA and Game24.

# D. Additional Empirical Verification

### D.1. Snowball Error Verifications on Larger LLMs

For a more robust verification of our theoretical findings, we have conducted extensive analyses on larger language models (Qwen2.5-14B-Instruct and Qwen2.5-32B-Instruct) in addition to the 7B/8B models presented in Figure 2. We maintained identical experimental settings and workflow as described in the original experiments to ensure methodological consistency, and we present the additional results in Figure 4.

Our key findings from these additional experiments demonstrate that the MI decay pattern remains consistent across larger model sizes, exhibiting similar behavior to smaller models. Also, response quality continues to show a negative correlation with output length, as observed in our original experiments.

### D.2. Snowball Error Analysis by Difficulty Levels

We have conducted verifications for the snowball errors at different difficulty levels. The experiments are performed on MATH-500 (Lightman et al., 2023), where the questions are divided into 5 different difficulty levels (level 1 for the simplest, level 5 for the most difficult). The results are presented in Figure 5. In (a), we present the relationship between the estimated MI and average response length at different levels. For all 5 levels, we observe similar MI decay phenomena like Figure 2 in our main text. In (b), we present the relationship between the average accuracy and the length of responses at different levels. For all 5 levels, we observed that accuracy generally decreases with response length. Furthermore, the accuracy of the simplest level 1 decreases significantly when the length increases, demonstrating the harm of overthinking. These results demonstrate that the MI decay phenomenon is highly relevant to the response length even when the influence of question difficulty is similar.

### D.3. Influence of $k$

For better understanding Theorem 4.6, we have conducted an analysis about the influence of $k$ in the theoretical findings, which is presented in Figure 6. The experiments are performed on GSM8k and PrOntoQA.

The results illustrate that: (1) the reasoning correctness increases when the reasoning cost $k$ increases, which aligns with the theoretical findings in Theorem 4.6; (2) for simpler PrOntoQA task where the value function is more reliable, the accuracy

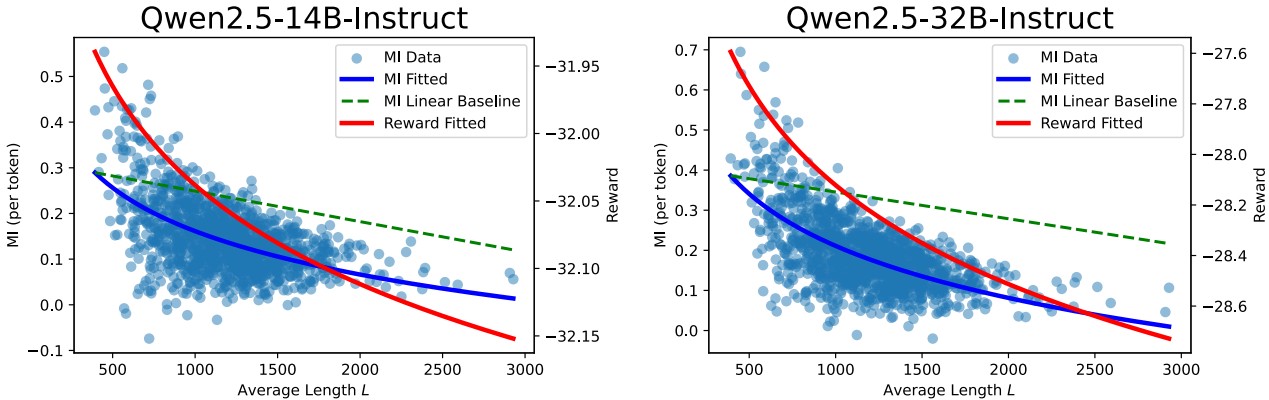

*Figure 4.* Additional snowball error verifications. We have conducted extra experiments on larger LLMs, including Qwen2.5-14B-Instruct and Qwen2.5-32B-Instruct, with the same setting as results in Figure 2 of our main text.

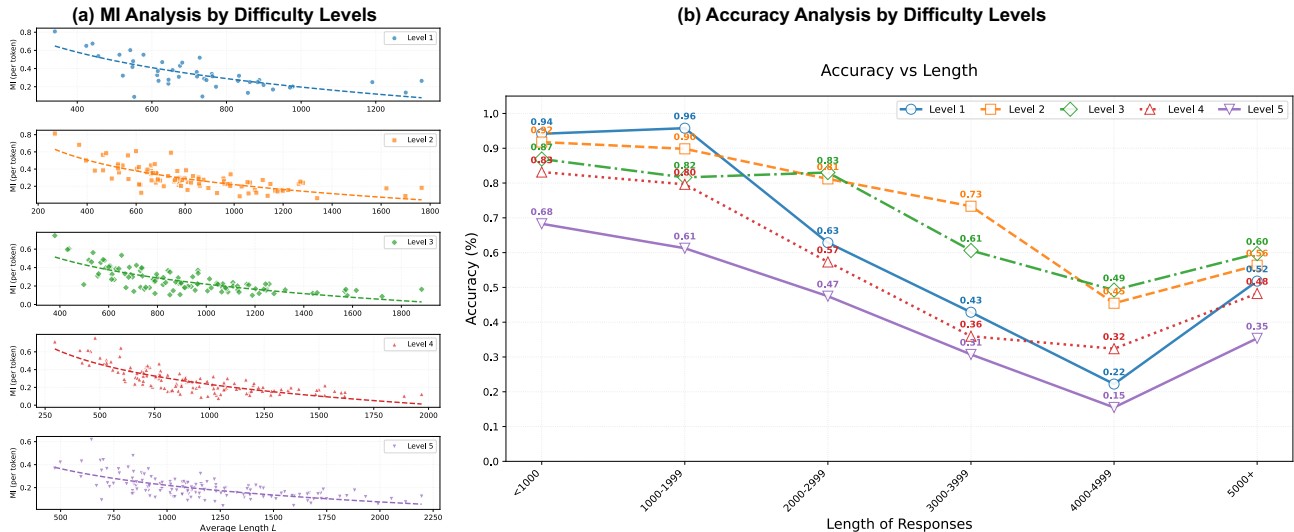

*Figure 5.* Snowball error analysis by different difficulty levels. We have conducted extra experiments to analyze the mutual information (MI) and accuracy at different difficulty levels. The questions at the same level share similar difficulty. (a) The MI analysis by difficulty levels. (b) The accuracy analysis by different difficulty levels.

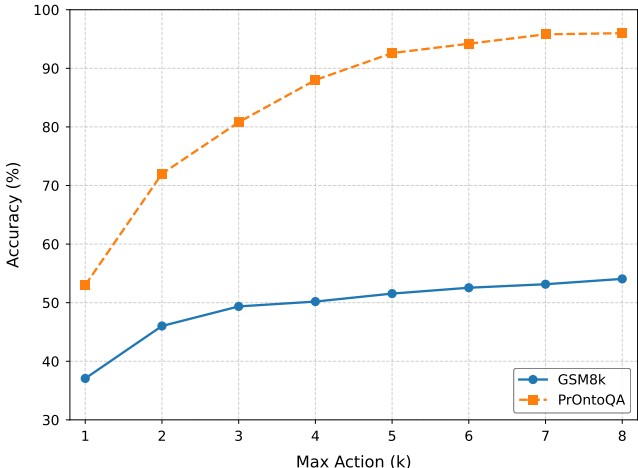

*Figure 6.* Influence of $k$. We have conducted extra experiments to further verify Theorem 4.6 by examining the influence of $k$. The experiments are performed on GSM8k and PrOntoQA, and we vary the value of the max-action of an MCTS while keeping other hyperparameters optimal, according to previous studies.

| $N$ | 2 | 4 | 6 | 8 | 10 | 12 | 14 | 16 | 18 | 20 |
|---|---|---|---|---|---|---|---|---|---|---|
| **ORM-Vote** | 17.96% | 29.01% | 38.67% | 46.13% | 50.83% | 53.87% | 59.94% | 63.81% | 66.85% | 67.68% |
| **ORM-Max** | 17.96% | 29.01% | 38.67% | 46.13% | 50.83% | 53.87% | 59.94% | 63.81% | 67.13% | 68.23% |

*Table 3.* Additional verification for BoN v.s. MCTS in Game24. The baseline MCTS obtain an accuracy of 64.80% where $\widetilde{N}_{\mathrm{res}} = 6.08$ and $\widetilde{N}_{\mathrm{call}} = 18.24$.

increases faster than GSM8k when $k$ increases.

### D.4. Additional Verification for BoN v.s. MCTS

We conducted additional verification experiments using the Game24 benchmark (Yao et al., 2024), which requires solving arithmetic puzzles to obtain the number 24. The setting is similar with Figure 3, and we present the results in Table 3.

This result illustrates a similar conclusion to Fig.3, when the total reasoning cost is comparable, BoN can achieve a comparable and even better performance than MCTS.

## E. More External Slow-Thinking Mechanisms Analysis

The external slow-thinking methods sometimes not only expand the width of the search space, but also perform repeatedly sampling and evaluating.

For example, besides expanding the width, Lookahead Search increase the computation steps in evaluating the thought by looking ahead for $r$ steps. This can be seen as a test of whether the current thought is good enough, i.e., affecting $\epsilon_b$.

An intuitive understanding is that the more steps we look ahead, the more likely we can find the best thought. This is due to the fact that the probability of generating a correct thought is related to the layer index $l$, and the probability of generating a correct thought increases with the layer index $l$.

**Definition E.1.** ($\delta$-wrong thought.) A thought $r_l$ is considered as $\delta$-wrong if the difference between the thought and the golden thought is larger than $\delta$, i.e., $|\phi(r_l) - \phi(r_l^*)| \geq \delta$.

**Lemma E.2.** *(probability of generating $\delta$-wrong thought.) The probability of generating a $\delta$-wrong thought is:*

$$\Pr\left[|\phi(r_l) - \phi(r_l^*)| \geq \delta\right] = \max(1 - \lambda_\delta e^{-l}, 0). \tag{20}$$

**Definition E.3.** (distinguishable condition.) We now assume that $\epsilon_b$ is guaranteed only if the correct thought and woring thuought are distinguishable. That is, given a correct thought $r_l$ (should be selected) and a wrong thought $r_l^-$ (should not be selected), the following inequality holds:

$$\Pr\left[\phi(r_l) - \phi(r_l^-) \geq \delta - \tau\right]. \tag{21}$$

We further analysis the lower bound of eq. (21) in the following theorem.

**Theorem E.4.** *(lower bound of distinguishable condition.) The distinguishable condition is guaranteed with the following probability lower bound with the condition when $l \geq \ln \lambda_\tau$:*

$$\Pr\left[\phi(r_l) - \phi(r_l^-) \geq \delta - \tau\right] \geq \lambda_\tau e^{-l} - \lambda_\tau \lambda_\delta e^{-2l}. \tag{22}$$

*Proof.*

$$\begin{aligned}
\Pr\left[\phi(r_l) - \phi(r_l^-) \geq \delta - \tau\right] &= \Pr\left[\phi(r_l^*) - \phi(r_l^-) - [\phi(r_l^*) - \phi(r_l)] \geq \delta - \tau\right] \\
&\geq \Pr\left[\phi(r_l^*) - \phi(r_l^-) \geq \delta\right] \times \Pr\left[\phi(r_l^*) - \phi(r_l) \leq \tau\right].
\end{aligned} \tag{23}$$

For the first term, since $\Pr\left[\phi(r_l^*) - \phi(r_l^-) \geq \delta\right] = \max(1 - \lambda_\delta e^{-l}, 0)$, we have:

$$\Pr\left[\phi(r_l^*) - \phi(r_l^-) \geq \delta\right] \geq 1 - \lambda_\delta e^{-l}. \tag{24}$$

While for the second term, since $l \geq \ln \lambda_\tau$, we have $\lambda_\tau e^{-l} \leq 1$, thus

$$\Pr\left[\phi(r_l^*) - \phi(r_l) \leq \tau\right] = \min(\lambda_\tau e^{-l}, 1) = \lambda_\tau e^{-l}, \tag{25}$$

With eq (23), we have

$$\begin{aligned}
\Pr\left[\phi(r_l) - \phi(r_l^-) \geq \delta - \tau\right] &\geq \Pr\left[\phi(r_l^*) - \phi(r_l^-) \geq \delta\right] \times \Pr\left[\phi(r_l^*) - \phi(r_l) \leq \tau\right] \\
&\geq (1 - \lambda_\delta e^{-l}) \times \lambda_\tau e^{-l} \\
&= \lambda_\tau e^{-l} - \lambda_\tau \lambda_\delta e^{-2l}.
\end{aligned} \tag{26}$$

This finishes the proof.

$\square$

With theorem E.4, we can derive the best setting of rollout step $\gamma$ in Lookahead Search.

**Corollary E.5.** *(best setting of rollout step $\gamma$ in Lookahead Search.) The best setting of rollout step $\gamma$ in the $l$-th layer in Lookahead Search is:*

$$\gamma_l^* = \max(\ln 2\lambda_\delta - l, 0). \tag{27}$$

*Proof.* Evaluating the thought in the $l$-th layer by looking ahead for $\gamma$ steps, the lower bound of distinguishable condition is:

$$f(\gamma_l) = \lambda_\tau e^{-(l+\gamma_l)} - \lambda_\tau \lambda_\delta e^{-2(l+\gamma_l)}. \tag{28}$$

the derivation of $f(\gamma_l)$ is:

$$\begin{aligned}
\frac{\mathrm{d}}{\mathrm{d}\gamma_l} f(\gamma_l) &= -\lambda_\tau e^{-(l+\gamma_l)} + 2\lambda_\tau \lambda_\delta e^{-2(l+\gamma_l)} \\
&= -\lambda_\tau e^{-(l+\gamma_l)}(1 - 2\lambda_\delta e^{-l}).
\end{aligned} \tag{29}$$

Let $\frac{\mathrm{d}}{\mathrm{d}\gamma_l} f(\gamma_l) = 0$, we have:

$$\tilde{\gamma}_l = \ln 2\lambda_\delta - l. \tag{30}$$

Considering the practical meaning, we have

$$\gamma_l^* = \max(\tilde{\gamma}_l, 0) = \max(\ln 2\lambda_\delta - l, 0). \tag{31}$$

This finishes the proof. $\square$

This corollary shows that the best setting of rollout step $\gamma$ in Lookahead Search is related to the layer index $l$, and not always the larger the better.

