# OpenReview forum: "Rethinking External Slow-Thinking: From Snowball Errors to Probability of Correct Reasoning"
_ICML.cc/2025/Conference — ICML 2025 poster_

### Official Review · Reviewer_S59i · 2025-03-16

**Overall Recommendation:** 3

**Summary:**

The paper provides theoretical analysis of the snowball error in reasoning models with external thinking mechanism. The paper unveil some interesting properties of reasoning models. However, the strong assumptions make most results somewhat obvious and/or unhelpful.

**Claims And Evidence:**

- The setting in the paper is oversimplified, leading to loose bounds and somewhat obvious/unhelpful results.
    - The authors did not discuss the possibility that the thoughts are incorrect. They only analyzed the errors in transforming thoughts into reasoning tokens. The paper, therefore, provides an incomplete picture of the problem (and a complete analysis is what I expected after reading the introduction. IMHO, the author should rewrite parts of the paper to better reflect the contributions).
    - The facts that thoughts and reasoning steps are generated sequentially and one mistake in the sequence can lead to wrong output were not discussed and ignored in the formulation.  In the paper, the accumulative error is simply the sum of errors at all steps.
    - The gap between the true error and the upper bound in theorem 3.3 is too big (Fig. 2). This could be a consequence of the weakness mentioned in the previous point.
    - Due to the strong assumption on $P(e)$, theorem 4.4 is not very informative. The probability of correct answers decays too fast at the rate of $\Theta(e^{-L^2})$. This goes against empirical evidence where models can still generate long correct answers with decent probability.

Post rebuttal comments: I raised the score to 3.

**Essential References Not Discussed:**

No

**Experimental Designs Or Analyses:**

Yes.

**Methods And Evaluation Criteria:**

- This is a theory paper so the experiments are for demonstration purposes. The presented experiments are good but some more experiments are needed to validate the theory. For example, experiments for theorem 4.4 and lemma 5.x are missing.

**Other Comments Or Suggestions:**

No.

**Other Strengths And Weaknesses:**

No.

**Questions For Authors:**

Please address the weaknesses mentioned in the previous sections.

**Relation To Broader Scientific Literature:**

The paper can have a good contribution to the literature if the weaknesses are addressed. Understanding the mechanisms for reasoning is important as it can lead to more effective and efficient reasoning methods.

**Theoretical Claims:**

As stated in the Claim section, the theory is based on strong assumptions leading to somewhat obvious and unhelpful conclusions.

---

> ### Author Rebuttal · Authors · 2025-03-30
>
> We sincerely appreciate your valuable feedback. Below are our responses to the concerns raised.
>
> ## About the Probability that the Thoughts are Incorrect
> Our theoretical analysis hinges on the framework presented in Fig.1, where we model the LLM reasoning process as a *planning-execution* mechanism. Given a question, the LLM first plans potential reasoning paths, generating implicit thoughts ($t$), before executing the tasks in these thoughts to produce responses ($r$).
>
> We primarily focus on error propagation in the *execution* phase, which is justified because **correct planning is a prerequisite for correct reasoning**—planning errors inherently lead to incorrect results. **However, empirical evidence suggests that even highly capable models can still commit errors during execution**, which is our primary analytical focus.
>
> We will improve presentation clarity in the revision.
>
> ## About the Formulation of Intermediate Error
> We explicitly recognize that *a single mistake in the sequence can lead to incorrect output*, which is naturally included in our setting. Our formulation captures how snowball errors accumulate across reasoning steps ($l$), ultimately exceeding a threshold and causing reasoning failure. **In other words, snowball errors accumulate incrementally, while reasoning errors directly emerge from excessive accumulation.**
>
> Our formulation ***does*** consider the influence of prior steps. **Step $l$ in our framework accounts for prior influences through prior snowball errors (or information loss)**, as defined in Definition 2.2. Furthermore, Theorem 3.3 evaluates reasoning error probability at each step $l$, **explicitly linking it to accumulated errors from previous steps**. Thus, **our analysis operates at a step-level granularity, not merely aggregating errors over all steps.**
>
> ## About the Results of Theorem 3.3
> Theorem 3.3 actually establishes a ***lower bound*** for reasoning error probability $e_l$ at step $l$. As snowball errors accumulate with increasing $l$, the error probability rises accordingly.
>
> This result is empirically verified by Fig.2 from two aspects.
> - Mutual information decreases (snowball error increases) with $l$ (Definition 2.1 & 2.2).
> - Response quality degrades with decreased MI.
>
> Additional experimental verification is available in our anonymous repository: https://anonymous.4open.science/r/extra-experiments-0E75/README.md, we kindly refer you to the ***Analysis by Difficulty*** section, also in our discussion with reviewer Qdus (section "The Snowball Error and the Length"). We will incorporate these supplemental results in our revision.
>
> ## About the Results of Lemma 4.4
> We understand your concerns may be about ***lemma 4.4*** and will respond from the following two aspects:
> 1) **The assumptions.** Proposition 4.3 facilitates clarity and subsequent analyses (Sec 4.2, line 214). Our results generalize to any scenario where $\operatorname{Pr}\left[|\phi(r_l)-\phi(r_l^*)| \leq \tau\right]$ decreases monotonically with $l$. We kindly refer you to our discussion with reviewer yYwM (section "Lack of Empirical Verification for Proposition 4.3").
> 2) **Further illustration for Lemma 4.4.** The "$\leq$" in 4.4 originates from the operator "$\operatorname{min}$" in Proposition 4.3, where the single-step correct reasoning probability is controlled by a constant $\lambda_\tau$. While initially loose for small $l$, **the bound tightens as $l$ increases**, especially when $\lambda_\tau e^{-l} < 1$, **reflecting rapidly increasing error probabilities in later reasoning steps.**
>
> ## Experiments for Theorem 4.6
> We take theorem 4.6, which is a main result in our paper and lemma 5.x are its derivations, for the subsequent response.
>
> Theorem 4.6 relates reasoning cost ($k$), length ($L$), and value function reliability ($\epsilon_b$) to the upper bound of correct reasoning probability.
>
> **For $L$:**
>
> The impact of reasoning length has been implicitly verified through:
> - ***Analysis by Difficulty*** section in our repository.
> - Fig. 2 results.
>
> This evidence confirms that longer paths increase errors.
>
> **For $k$ and $\epsilon_b$:**
>
> We empirically validate $k$'s impact through RAP-like MCTS experiments [1]. See section ***Influence of $k$*** in our repository. Results show improved reasoning accuracy with increased $k$. And the $\epsilon_b$ effect manifests in PrOntoQA's faster accuracy improvement versus GSM8k due to its simpler nature.
>
> Besides, many external slow-thinking methods (listed in our related works) have empirically verified that increasing the cost in inference time can improve the model's reasoning capability, a.k.a. ***test-time scaling laws***.
>
> [1] AlphaZero-like tree-search can guide large language model decoding and training (ICML 2024).
>
> While simplifying CoT formulations, our work provides essential foundations for execution-error analysis—a crucial first step acknowledged by other reviewers. We appreciate your recognition of this focused contribution.

---

> > ### Comment · Reviewer_S59i · 2025-04-08
> >
> > Thank you for your rebuttal. Although the rebuttal provided some clarifications, I still have concerns regarding the assumptions. I think it is hard to get tighter bounds or generalize the results to more realistic settings with the current assumptions. If the authors could outline a plan to improve the paper in the future, I will raise my score.

---

> > > ### Author Response · Authors · 2025-04-08
> > >
> > > We sincerely appreciate your comments and valuable suggestions.
> > >
> > > We fully acknowledge the inherent challenges in formalizing CoT reasoning given its complexity and abstract nature in LLMs. As Reviewer NREt noted, while existing research primarily focuses on empirical evaluations, they cannot *formally explain why slow-thinking methods work or when they fail.* Our work represents a concerted effort to ***"bridge the gap between empirical success of slow-thinking methods and their theoretical understanding".***
> > >
> > > We agree that achieving fully explainable LLM reasoning remains an ambitious long-term goal for the research community. **While our current work provides foundational theoretical insights, we recognize it as an initial yet significant step toward this broader vision.** We anticipate that subsequent research—both our ongoing studies and future work by the community—will refine and expand upon these foundations. Ultimately, this iterative progress will lead us closer to truly interpretable reasoning in LLMs.
> > >
> > > In response to the reviewers' suggestions, we propose concrete plans for both **immediate improvements** to our current manuscript and **future research** directions:
> > >
> > > **(1) Immediate Improvements to Our Paper**
> > >
> > > > Plan 1: Improving the Validity of Theoretical Assumptions
> > >
> > > As our response to Reviewer yYwM, we can relax Proposition 4.3's conditions to require only a monotonic decrease rather than the original exponential form. This relaxation maintains theoretical robustness while yielding more general results:
> > > the upper bound of lemma 4.4 and theorem 4.6 will be derived as $\xi^L(l,\tau)$ and $\epsilon_b^L k^L \xi^L(l,\tau)$ respectively, where $\xi^L(l,\tau):=\prod_{l=1}^{L}\xi(l,\tau)$.
> > > This modification ensures broader applicability by weakening assumptions on $P(e)$, addressing concerns regarding the original constraints. We will formally include this improvement in our revision.
> > >
> > > > Plan 2: Clarifying Sequential Reasoning Formulation
> > >
> > > Critical to our formulation is the principle that *a single mistake in reasoning can lead to incorrect final outputs*—even if the final answer appears correct by coincidence. Specifically, we implicitly assume that **any intermediate mistake compromises reasoning validity.**
> > >
> > > Building upon this idea, Secs 2 & 3 analyze how the probability of error at step relates to reasoning depth $l$ (e.g., Theorem 3.3’s lower bound). This result is just designed to characterize the error probability of **every intermediate reasoning step**. In our revision, we will explicitly articulate this motivation and its implications, ensuring clearer presentation.
> > >
> > > > Plan 3: Incorporating Error Probability Analysis for Thoughts
> > >
> > > Our theoretical framework conceptualizes reasoning as a planning-execution process, where correct planning (thought generation) is foundational for accurate reasoning.
> > >
> > > To further analyze error propagation, we will incorporate a detailed discussion on the thought errors. Specifically, we will establish a sequential error propagation framework similar to [1], $\forall l \leq L, $ we will model the relationship between thought error probability and factors like model capability and reasoning depth $l$, i.e. $P_{err}(t_l) = \epsilon(\pi,l)$. This extension will enhance the practical validity and completeness of our theoretical framework.
> > >
> > > [1] The Pitfalls of Next-Token Prediction. ICML 2024.
> > >
> > > > Plan 4: Enhanced Analysis of Mutual Information (MI) Decay
> > >
> > > To strengthen our discussion on MI decay (assumption in Lemma 3.2), we will provide a more rigorous justification for why task-relevant MI decreases with reasoning depth $l$ or how earlier steps influence information loss in later steps.
> > >
> > > This premise has already been empirically supported by the results in Fig.2. Theoretically, we argue that, since $r_l = \pi(r_{<l})$,
> > > if $\pi$ introduces no additional task-relevant information, the relevance of $r_l$ cannot exceed that of $r_{<l}$, leading to MI decay. A more detailed formulation will be included in the revision.
> > >
> > > > Plan 5: Expanding Empirical Validation
> > >
> > > Due to time and resource constraints during rebuttal, comprehensive empirical validation in real-world settings is difficult. However, in our revision, we plan to:
> > > - Test our theoretical predictions across a broader range of hyperparameters (e.g., MCTS configurations).
> > > - Validate findings on additional search algorithms, benchmarks, and larger LLMs.
> > >
> > > **(2) Future Research Directions**
> > >
> > > 1) **Reflection Mechanism.** We will investigate how reflection can act as an error-reset mechanism with potential trade-offs in efficiency.
> > >
> > > 2) **Implicit Reasoning.** Our results may extend to implicit reasoning by formalizing error propagation in looped structures.
> > >
> > > Due to the character limit, we kindly refer you to our discussion with reviewer Qdus for more details.
> > >
> > >
> > > Your insightful advice is invaluable in shaping this work, and **we would be most grateful if you could consider raising your score for our work!**

---

### Official Review · Reviewer_NREt · 2025-03-17

**Overall Recommendation:** 4

**Summary:**

This paper discusses the issue of understanding and improving external slow-thinking methods in LLM reasoning. The authors (1) propose a theoretical framework based on information theory that analyzes snowball errors and (2) connects them to the probability of reasoning errors in LLMs. This method aims to explain why external slow-thinking works, quantify its limitations, and inform better design of slow-thinking strategies over baseline methods. They provide experimental results to demonstrate how mutual information decays along reasoning steps and validate the presence of snowball errors. They also offer a theoretical comparison of different external slow thinking methods.

**Claims And Evidence:**

Most of the paper’s claims are supported by empirical evidence. The most important observations are that mutual information tends to decay along reasoning steps and that snowball errors can occur in LLM reasoning chains.
I have concerns regarding some claims: (1) the general linkage between MI decay and reasoning errors. While MI decay may correlate with reasoning challenges, it is unclear that this directly causes errors, especially given that LLMs often exhibit self-correction abilities during generation. (2) the universality of snowball error dynamics across all slow-thinking methods. Best-of-N and MCTS have fundamentally different mechanisms for handling reasoning. Simply applying a single information-theoretic framework may oversimplify these differences.

**Essential References Not Discussed:**

Not found.

**Experimental Designs Or Analyses:**

The experiments are extensive, with detailed analysis of the results. These experiments validate the effectiveness of the proposed information-theoretic framework in capturing mutual information decay and reasoning errors. The authors also demonstrate the method's robustness and generality across different sizes and complexities of reasoning datasets and baselines.

**Methods And Evaluation Criteria:**

The authors use widely adopted benchmark datasets that are appropriate for evaluation.

**Other Comments Or Suggestions:**

Adding a more detailed explanation of Figure 2, including how to interpret the MI curves in relation to reasoning quality, would improve clarity.

**Other Strengths And Weaknesses:**

See above.

**Questions For Authors:**

See above.

**Relation To Broader Scientific Literature:**

LLM reasoning and CoT prompting. This paper offers a new theoretical perspective on error accumulation.
Best-of-N and MCTS. This paper provides an information-theoretic framework to analyze their limitations.
Error propagation. This paper formalizes these as snowball errors driven by mutual information decay.

**Theoretical Claims:**

The idea of analyzing and improving external slow-thinking in LLM reasoning through an information-theoretic framework is novel and timely. Most existing work focuses on empirical evaluations of prompting or decoding strategies, but these approaches cannot formally explain why slow-thinking methods work or when they fail, limiting our theoretical understanding and principled design of such methods. This paper introduces a method based on mutual information decay analysis to model snowball errors in reasoning, as well as significantly improves our understanding of the challenges and trade-offs in multi-step LLM reasoning.

The workflow is well-structured, as it connects theoretical modeling with empirical validation on standard reasoning benchmarks to make the analysis both grounded and actionable. This approach bridges the gap between empirical success of slow-thinking methods and their theoretical understanding, and further provides guidance for designing more robust reasoning strategies in LLMs.

---

> ### Author Rebuttal · Authors · 2025-03-30
>
> We sincerely appreciate the insightful and constructive feedback provided in your reviews. Below, we address each concern in detail.
>
> ## The General Linkage between MI Decay and Reasoning Errors
> We formalize the reasoning process of LLMs as a *planning-execution* framework. Given a question, as illustrated in Fig.1, an LLM first plans a potential reasoning path, thereby generating implicit thoughts ($t$). Subsequently, the LLM attempts to execute the tasks specified in these thoughts, producing the observable responses ($r$).
>
> In this work, our primary focus is analyzing the probability of correct reasoning, particularly examining error propagation in the *execution* phase under **external search**—where search algorithms are guided by external mechanisms (e.g., an auxiliary value model). By contrast, the "self-correction" capability (or reflection) primarily arises from the model’s **internal reasoning ability**. Specifically, the model autonomously detects potential errors and restarts the execution process from an earlier step.
>
> **The internal reflection mechanism constitutes the key distinction between external and internal slow-thinking methods.**
> - External slow-thinking methods rely on **width-expansion**, broadening the search space.
> - Internal slow-thinking methods leverage **reflection**, enabling recovery from intermediate states.
>
> We intend to conduct further analyses on the reflection mechanism in future work, where **it can be rigorously modeled as a form of error accumulation with inherent trade-offs in error detection**. Additional discussion on this topic can be found in our response to Reviewer Qdus (Section: Extension to AoT and AoT+).
>
> Additionally, extra experiments have been conducted to verify the relationship between the MI, accuracy and lengths. The results are presented through an anonymized repository (https://anonymous.4open.science/r/extra-experiments-0E75/README.md), the ***Analysis by Difficulty*** section. We also kindly refer you to our discussion with reviewer Qdus (section "The Snowball Error and the Length") for contexts of this additional verification.
>
> We thank you for your valuable suggestions and will enhance the clarity and comprehensiveness of our manuscript in the revised version.
>
> ## The Universality of Snowball Error Dynamics across All Slow-Thinking Methods
> As discussed in the previous section, while Best-of-N (BoN) and Monte Carlo Tree Search (MCTS) employ distinct search strategies, they share a fundamental similarity: **both expand the search space under external value guidance.** Consequently, they qualify as external slow-thinking methods, aligning with our analytical framework. From this perspective, any search strategy based on width expansion falls within the scope of our theoretical results.
>
> Moreover, we believe our findings can extend to other slow-thinking paradigms, including **internal slow-thinking** (reflection-based) and **implicit reasoning** methods. For a more detailed discussion, please refer to our response to Reviewer Qdus (Section: Extension for More Test-Time Scaling Methods).
>
> ## Detailed Explanation of Figure 2
> We sincerely appreciate your thorough evaluation of Figure 2. Below, we simply summarize the key experimental settings (additional details will be provided in the appendix in the revised version):
>
> 1) **Prompting & Inference:**
>     - Models are prompted (see Appendix B.1) to generate step-by-step reasoning.
>     - Observed answers ($r$) and implicit thoughts ($t$, derived from ground-truth rewrites) are collected.
> 2) **Mutual Information (MI) Estimation:**
>     - Following prior work, we compute and estimate the MI between $r$ and $t$.
>     - Each question’s result is plotted as a blue dot in Figure 2.
> 3) **Reasoning Quality Evaluation:**
>     - An outcome reward model assesses reasoning correctness.
>     - The relationship between MI and reasoning quality is fitted to produce the final curve in Figure 2.
>
> In the camera-ready version, we will elaborate on Appendix B.1 as per your suggestion to improve clarity.

---

> > ### Comment · Reviewer_NREt · 2025-04-04
> >
> > Thank you for clarifying. I will keep the score.

---

> > > ### Author Response · Authors · 2025-04-07
> > >
> > > We are deeply grateful for your steadfast support and valuable suggestions for our research.

---

### Official Review · Reviewer_A7PQ · 2025-03-17

**Overall Recommendation:** 1

**Summary:**

This paper analyzes the potential snowball error effect that may occur during the reasoning process of Large Language Models (LLMs), and connects it to the probability of correct reasoning using information theory. Within this theoretical framework, external slow thinking methods can be interpreted as strategies for reducing the probability of errors.

**Claims And Evidence:**

Yes.

**Essential References Not Discussed:**

N/A

**Experimental Designs Or Analyses:**

The paper conducts experimental validation for its two proposed theoretical insights, making them both reliable and clear.

**Methods And Evaluation Criteria:**

Yes.

**Other Comments Or Suggestions:**

N/A

**Other Strengths And Weaknesses:**

Strengths：
1.The paper provides a very clear introduction to the background and its proposed theoretical method.
2.The theoretical framework and contributions of the paper are substantial and meet the high standards expected for ICML submissions.

Weaknesses：
1. The LaTeX formatting of the paper contains errors, and the wrong template appears to have been selected
2. The choice of prior work to follow seems questionable; please refer to the section on theoretical claims for specific concerns.
3. The two main aspects of the paper do not seem to form a cohesive synergy. Specifically, I find that the connection between snowballing errors
 and overthinking is not very apparent.
I would suggest that the authors provide a more intuitive explanation, rather than relying solely on theoretical arguments for acceptance.

**Questions For Authors:**

Please refer to the section on theoretical claims and Weaknesses.

**Relation To Broader Scientific Literature:**

Please refer to the section on theoretical claims.

**Theoretical Claims:**

In fact, I have thoroughly studied and read the work which is followed by this paper: "Understanding chain-of-thought in LLMs through information theory." This work proposes a theoretical framework for evaluating the correctness of each step in the chain-of-thought (COT) using information theory. However, I personally believe that this work has two major issues ：it has not yet been published or widely validated and recognized. I would like to ask the authors whether they have conducted an analysis of the shortcomings of this theory and made improvements, or if they have directly followed the work as it is.

---

> ### Author Rebuttal · Authors · 2025-03-30
>
> We sincerely appreciate your valuable feedback. Below, we address your concerns point by point.
>
> ## Regarding Prior Work [1]
> While our work is inspired by [1], and both employ information theory to analyze chain-of-thought (CoT) reasoning, **there are significant distinctions between our approaches.**
>
> The primary focus of [1] aligns with process reward models (PRMs) [2], which seek to evaluate the quality of intermediate reasoning steps. To achieve this, [1] leverages information-theoretic metrics to estimate information gain as supervisory signals. **In contrast, our work aims to provide a theoretical foundation for understanding the *mechanism* of multi-step reasoning in LLMs.**
>
> We consider the former peer reviews of [1] for further discussion. The major concerns raised in [1]’s ICLR 2025 rebuttal (https://openreview.net/forum?id=ouRX6A8RQJ) primarily revolve around experimental design, whereas its theoretical formulation was generally praised. **From the theoretical perspective, our paper is not a direct extension of [1]**; a potential overlap lies in modeling CoT as a task-execution process, but our formulations differ substantially.
>
> Specifically, we overcome several limitations of [1]:
>
> - [1] relies on complex concepts (e.g., primitive tasks) and assumes identifiable tasks as combinations thereof. In contrast, we formulate reasoning as a transparent planning-execution process, bridging implicit thoughts to observable responses.
> - [1] introduces additional assumptions (e.g., Bayesian networks) to model reasoning errors, whereas our framework is derived from simpler formulation and information-theoretical foundations (e.g. Fano’s inequality), culminating in Theorem 3.3, which establishes a clear lower bound with fewer assumptions.
>
> **In summary, our work diverges from [1] in scope, reasoning formulation, and error modeling. Although both studies build upon information theory, we argue that they contribute distinct perspectives.** We hope this clarification enhances the understanding of our paper’s novel contributions.
>
> [1] Ton, J. F., Taufiq, M. F., & Liu, Y. (2024). Understanding Chain-of-Thought in LLMs through Information Theory. arXiv preprint arXiv:2411.11984.
>
> [2] Lightman, H., Kosaraju, V., Burda, Y., Edwards, H., Baker, B., Lee, T., ... & Cobbe, K. (2023, May). Let's verify step by step. In The Twelfth International Conference on Learning Representations.
>
> ## Regarding LaTeX Formatting
> We confirm that our manuscript adheres to ICML 2025’s LaTeX template (https://icml.cc/Conferences/2025/AuthorInstructions), ensuring compliance with formatting guidelines.
>
> ## Clarifying the Paper’s Structure
> Thank you for your inquiry about the connection between the two aspects of our paper. Below, we provide a detailed explanation:
>
> Overall, the structure of our paper can be divided into 2 parts:
> 1) **Part 1 (Secs. 2–3):** We analyze "snowball errors" and their relationship with reasoning errors. Theorem 3.3 formalizes this, with $H_{<l}(t|r)$ quantifying error accumulation—analogous to a snowball growing in size.
> 2) **Part 2 (Secs. 4–6):** Leveraging reasoning error probabilities, we theoretically analyze scaling methods (Theorem 4.6, Table 1, Figure 3). As noted in Line 270, correct reasoning is framed probabilistically (generation + selection), with external strategies (e.g., width expansion) aligning with variables $k$ and $b$.
>
> **For Part 1:** The key theoretical insight (Theorem 3.3) formalizes "snowball errors"—minor inaccuracies in early reasoning steps that amplify over subsequent steps, analogous to how a snowball grows when rolled. This is quantified via conditional entropy $H_{<l}(t|r)$, which captures the compounding effect of errors. An intuitive analogy is provided in Sec.2 (~Line 104).
>
> **For Part 2:** Theorem 4.6, Table 1, and Figure 3 analyze test-time scaling methods. We frame correct reasoning probabilistically, decomposing it into generation (producing candidate solutions) and selection (identifying the optimal one), as elaborated in Line 270. External slow-thinking methods relying on width expansion align with our theoretical variables $k$ and $b$, offering a principled justification for their efficacy.
>
> For better illustration, We are happy to incorporate these intuitive explanations more prominently in the revised manuscript to improve clarity. **We also hope our response can address your misunderstanding of our works and reconsider our contributions!**

---

> > ### Comment · Reviewer_A7PQ · 2025-04-03
> >
> > Regarding the LaTeX Format: As far as I know, the ICML 2025 LaTeX template includes two versions: one for submission and another for the camera-ready version. By default, it uses the camera-ready format. It seems that your submission uses the camera-ready version. If I am mistaken, please feel free to correct me. Thank you.
> >
> > Regarding Prior Work: This submission received mostly positive scores in the ICLR review process, with a few negative ones. You mentioned that the main concerns centered around the experimental design, while the theoretical contributions were generally praised. However, this does not seem to be entirely accurate. While some positive reviews did indeed praise the theoretical aspects, they also criticized the experimental setup—specifically, the reliance on a single supervised model ( g ) to estimate mutual information. This limitation actually reflects a fundamental issue with the proposed method.
> >
> > In addition, many reviewers pointed out the questionable nature of the assumption regarding information gain. I believe this assumption is central to the use of information theory in addressing the Chain-of-Thought (CoT) problem. I would be interested to hear your thoughts on this assumption.
> >
> > I will carefully study your further responses and reply accordingly. If I have misunderstood anything, please do not hesitate to correct me.

---

> > > ### Author Response · Authors · 2025-04-03
> > >
> > > Thanks for your feedback. We are glad to further address your two concerns as follows.
> > >
> > > ## Response to LaTeX Format Concern
> > > While it is correct that the template includes both a submission version and a camera-ready version, we have verified that our submission strictly adheres to the submission version.
> > >
> > > As specified in the template provided in ICML author instructions (lines 21-25 of "example_paper.tex"), **the template version is determined by the package declaration:**
> > > - `\usepackage{icml2025}` denotes the double-blind submission version.
> > > - `\usepackage[accepted]{icml2025}` denotes the camera-ready version.
> > >
> > > **Our submission correctly employs the former package for double-blind review.** For further verification, we kindly invite you to:
> > > 1) Please verify that the reviewed PDF was correctly downloaded from OpenReview.
> > > 2) Compare our submission with ICML's official compiled templates.
> > > 3) Note the presence of left-aligned line numbers in our submission.
> > > 4) Confirm the proper anonymization of author information.
> > >
> > > These elements collectively demonstrate our compliance with the double-blind submission requirements.
> > >
> > > ## Regarding Prior Work [1]
> > > We appreciate the opportunity to clarify the relationship between our work and [1]. If we understand correctly, given that *[1] employs information theory to analyze CoT, and concerns have been raised regarding its information-theoretic results in its ICLR 2025 peer review*, your key question appears to be: ***Is information theory suitable for modeling CoT?***
> > >
> > > > **General Response:**
> > > >
> > > > Our work and [1] pursue **fundamentally different analysis objectives**, leading to **distinct formulations** of CoT. Therefore, the potential limitations of [1] do not apply to our contributions.
> > >
> > > Before elaborating, we highlight two key principles that guide our perspective:
> > > - Theoretical formulations serve as tools to address specific research questions.
> > > - Every theoretical framework has inherent limitations.
> > >
> > >
> > > **Our work differs substantially from [1] in research objectives:**
> > > - **[1]:** Uses information theory to introduce "information gain" as a metric for ***detecting reasoning errors.***
> > > - **Our Paper:** Leverages information theory to ***characterize the mechanisms underlying test-time scaling methods.***
> > >
> > > As a theoretical framework, **the suitability of information theory depends critically on the research objective.** While information theory offers intuitive and interpretable theoretical insights, **its quantitative applicability in natural language contexts (e.g., LLMs) is limited.**
> > >
> > > We have carefully examined [1]'s results. Although the left-hand side of Proposition 3.4 ("information gain" metric) provides an intuitive measure of step-wise significance, as noted by its reviewers, it faces two key limitations:
> > > - Its relevance to reasoning correctness remains debatable.
> > > - This metric cannot be calculated directly, thus a questionable supervisor model ($g$) needs to be introduced.
> > >
> > > However, **these limitations in [1] do not imply that information theory is unsuitable for CoT analysis.**
> > >
> > > **Our work avoids these pitfalls through a fundamentally different formulation, as explained in our initial rebuttal.**
> > > By modeling reasoning as a transparent planning-execution process linking implicit thoughts ($t$) to observable responses ($r$), we establish more reasonable assumptions (compared to [1]'s primitive tasks and Bayesian networks). Consequently, **Part 1 of our paper derives a lower bound on error occurrence probability** (Theorem 3.3) — a more robust result than [1]'s "information gain." Moreover, by focusing on trends in information loss (e.g., snowball effects), **we circumvent the need for exact quantification, thereby mitigating information theory's limitations.**
> > >
> > >
> > > In summary, our position is as follows:
> > > 1) [1]'s limitations do not invalidate information theory for CoT analysis.
> > > 2) [1]'s challenges stem from misalignment between its goals and information theory's constraints.
> > > 3) Our distinct objectives and formulation yield more reliable results while avoiding these pitfalls.
> > >
> > > We respectfully submit that our work should be evaluated independently of [1]'s potential shortcomings. We hope this response demonstrates that **information theory remains well-suited for our specific research objectives** in CoT analysis. **We would be deeply appreciative if you could consider raising your score of our work!**
> > >
> > > [1] Ton, J. F., Taufiq, M. F., & Liu, Y. (2024). Understanding Chain-of-Thought in LLMs through Information Theory. arXiv preprint arXiv:2411.11984.

---

### Official Review · Reviewer_Qdus · 2025-03-19

**Overall Recommendation:** 3

**Summary:**

The paper focuses on analyzing ``snowball effects'', where the model's implicit thinking process is not well represented by the tokens they generate at each step, hence accumulating throughout the inference. They analyze this phenomenon through information-theoretic perspective to give lower bounds for correct reasoning for well-known methods such BoN and ToT.

**Claims And Evidence:**

I found Figure 2 to be confusing. If the problem has more variables/steps to arrive at an answer, the length $L$ will naturally be greater, this might also reduce the performance since at each step, the model will need to choose the next step from more options (since we have more variables). However, this is also correlated with step length $L$, worrying me that the snowballing effect is a side effect of this, and not being able to represent the implicit thinking in the generated tokens. Perhaps, testing this idea on other datasets where the length of the solution does not necessarily imply a more difficult solution (such as adding $n$ many numbers, though this is just an option).

**Essential References Not Discussed:**

Not for the paper's results, but more discussions on test-time scaling methods would be helpful. Please see previous mentioned papers.

**Experimental Designs Or Analyses:**

Yes, see the previous section.

**Methods And Evaluation Criteria:**

See the previous section.

**Other Comments Or Suggestions:**

- Proposition 4.3 might have a typo in the results (see previous comment regarding this).

**Other Strengths And Weaknesses:**

Strengths:
- Easy to read and well-written
- The paper proposes a new information-theoretic perspective at investigating the behavior of various test-time scaling methods
- The topic is timely
- The experiments are mostly supportive of the narrative

Weaknesses:
- The algorithms of focus for the analysis can be increased to include methods that have a single context solution such as AoT, as opposed to ToT.
- Please see previous comments on Figure 2.

**Questions For Authors:**

- Do the authors think their analysis can be extended to more test-time scaling methods? And can be updated to include hints on the solution efficiency? (producing fewer tokens while having a similar performance)

In the current state of the paper, I give a ``Weak Accept'' rating, however, if authors make the necessary changes to improve the applicability of the paper as explained in my previous comments, I'd be happy to reconsider my score.

**Relation To Broader Scientific Literature:**

Test-time scaling is significant to improve the performance of pre-trained models in post-training phase, since the current models started to have slower improvements even though the training data and quality in the pre-training stage has never been higher. Having tools to analyze test-time scaling methods is important in the sense that researchers will have tools to improve upon these method. However, I believe the test-time scaling view in the paper, only represents a part of the literature. Newer test-time scaling methods such ``Algorithm of Thoughts'' (AoT) [1] and AoT+ [2] show improvements by scaling the reasoning within-context, in opposed to the methods only considering CoT chains in each context (such as BoN, ToT, GoT, RAP). I believe, if the paper can integrate their analysis to these cases, to perhaps explain their efficiency in terms of output tokens, it can be a significant addition to the reasoning/planning literature.

[1] Algorithm of thoughts: Enhancing exploration of ideas in large language models (ICML 2024), https://arxiv.org/abs/2308.10379
[2] LLMs Can Plan Only If We Tell Them (ICLR 2025), https://arxiv.org/abs/2501.13545

**Theoretical Claims:**

I did check the proofs. I believe there is a typo in the proposition 4.3, which the rhs should be 1 - rhs.

---

> ### Author Rebuttal · Authors · 2025-03-30
>
> We sincerely appreciate your valuable feedback and constructive comments. Below, we provide point-by-point responses to address the raised concerns.
>
> ## The Snowball Error and the Length
> We acknowledge your insightful observation that the length of responses could be correlated with question difficulty, beyond just snowball errors. This perspective significantly enriches the interpretation of our findings.
>
> To validate our claims in scenarios where solution length does not necessarily correlate with solution difficulty, **we have conducted additional experiments analyzing mutual information (MI) and accuracy across different difficulty levels**. These experiments were performed on the MATH-500 dataset [1], where questions are categorized into five difficulty levels (Level 1 being simplest, Level 5 most difficult), with questions at each level sharing comparable difficulty.
>
> The experimental results are available in our anonymized repository (https://anonymous.4open.science/r/extra-experiments-0E75/README.md), particularly in the ***Analysis by Difficulty*** section.
>
> Moreover, this finding aligns with recent literature. For instance, [2] demonstrates that excessively long CoT reasoning can harm solution accuracy, a phenomenon referred to as *overthinking* [3].
>
> [1] Lightman, H., Kosaraju, V., Burda, Y., Edwards, H., Baker, B., Lee, T., ... & Cobbe, K. (2023). Let's verify step by step. ICLR.
>
> [2] Wu, Y., Wang, Y., Du, T., Jegelka, S., & Wang, Y. (2025). When More is Less: Understanding Chain-of-Thought Length in LLMs. arXiv:2502.07266.
>
> [3] Sui, Y., et al. (2025). Stop Overthinking: A Survey on Efficient Reasoning for Large Language Models. arXiv:2503.16419.
>
>
>
> ## Extension to AoT and AoT+
> We appreciate the suggestion to explore extensions to AoT and AoT+. We would like to address this from two perspectives:
>
> 1) **External vs. Internal Slow-Thinking:** While AoT primarily utilizes ICL to teach algorithms to LLMs, **the introduction of specialized instances enables new capabilities like reflection**. This makes AoT more akin to internal slow-thinking approaches, whereas our current focus is on external slow-thinking methods.
>
> 2) **Reflection Mechanism:** However, **we are actively planning to incorporate reflection mechanisms in subsequent work**, as mentioned in later sections of this response. We believe this extension will substantially enhance the impact of our future research.
>
>
> ## Typos in Proposition 4.3
> After careful consideration, we believe Proposition 4.3 is mathematically sound.
>
> We would like to highlight Definition 4.1, where the left-hand item in Proposition 4.3 represents the probability of generating a ***correct*** reasoning step, which contrasts with Theorem 3.3 and the statements in line 213 (the probability of errors). We will improve the clarity of this presentation in our revised manuscript.
>
> ## Extension for more Test-Time Scaling Methods
> Our primary focus remains on external search methods, where we believe similar width-expansion strategies can be readily extended. Additionally, we have identified two other noteworthy test-time scaling approaches:
>
> 1) **Internal Slow-Thinking**. As introduced in our paper (Line 32), these methods employ training and parameter updates to enable long-CoT capabilities (e.g., DeepSeek-R1). **The key differentiator appears to be reflection mechanisms, allowing models to detect and correct reasoning errors.** By formalizing reflection, our framework could extend to understanding internal slow-thinking methods. One possible formulation can be built on the fact that ***reflection can reset the snowball error and thus obtain better results, with some trade-offs to detect the potential reasoning error***. This extension could significantly enhance our findings' broader implications.
> 2) **Implicit Reasoning**. This emerging paradigm [4-6] scales inference through looped computations rather than additional token generation. Our results may generalize to illustrate implicit reasoning benefits by formalizing error propagation in these looped structures.
>
> [4] Tack, J., et al. (2025). LLM Pretraining with Continuous Concepts. arXiv:2502.08524.
>
> [5] Chen, Y., et al. (2025). Inner thinking transformer: Leveraging dynamic depth scaling to foster adaptive internal thinking. arXiv:2502.13842.
>
> [6] Yu, Q., et al. (2025). Enhancing Auto-regressive Chain-of-Thought through Loop-Aligned Reasoning. arXiv:2502.08482.
>
> We believe these responses adequately address all concerns raised. We greatly appreciate your time and constructive feedback.

---

### Official Review · Reviewer_UzVe · 2025-03-20

**Overall Recommendation:** 3

**Summary:**

This paper aims to provide rationales for the effectiveness of inference-time compute scaling, also known as slow thinking, particularly from an information-theoretic perspective.

First, it argues that as the length of a reasoning path increases, the probability of encountering an error along the path also grows, potentially at a rate exceeding linear scaling.

Second, it posits that slow-thinking methods enhance reasoning by expanding the breadth of the reasoning space, thereby increasing the likelihood of generating a correct response. However, their effectiveness depends on the quality of the selection module, which is responsible for identifying the correct response from among the candidates.

All of these claims are substantiated and analyzed through mathematical frameworks and empirical experiments, at least according to the authors.

**Claims And Evidence:**

- Claim 1 (Section 3): The probability of a reasoning error is lower-bounded by a certain threshold, derived from the concept of snowball errors introduced in Section 2.
- Evidence 1: The mathematical proofs presented in Sections 2 and 3 appear reasonable, though I am not entirely certain. However, they rely on a strong assumption that each reasoning step follows a single gold-standard thought process. This assumption contradicts the widely accepted notion that multiple reasoning paths can lead to the same answer. Additionally, it is somewhat trivial to observe that as the length of a model-generated sequence increases, errors occurring earlier in the sequence are more likely to have a greater impact on later parts.

***

- Claim 2 (Section 4): The probability of correct reasoning in recent external slow-thinking methods depends on the combined process of generating multiple answer candidates and selecting the correct one from the pool.
- Evidence 2: This argument is reasonable; however, it is not entirely convincing to assume that there exists only a single gold-standard reasoning path, as stated in Section 4 (r_l^*).

**Essential References Not Discussed:**

Maybe not essential but related: How Language Model Hallucinations Can Snowball (ICML 2024)

**Experimental Designs Or Analyses:**

The empirical experiments are limited in terms of both the target tasks and the models used. Specifically, only 8B-scale models are evaluated, and the study focuses on just two tasks (GSM8K and ProntoQA). This raises concerns about the generalizability of the findings to models of different sizes and a broader range of tasks.

**Methods And Evaluation Criteria:**

The paper primarily focuses on investigating the inner workings of slow thinking rather than proposing a new solution or method for further improvement. As a result, it does not discuss the novelty or effectiveness of any proposed approach.

Nevertheless, the study would have been more insightful if it had incorporated a broader range of related methods beyond BoN and MCTS in its experiments.

**Other Comments Or Suggestions:**

Please see the above comments.

**Other Strengths And Weaknesses:**

Please see the above comments.

**Questions For Authors:**

Please see the above comments.

**Relation To Broader Scientific Literature:**

Inference-time compute scaling, also known as “slow thinking,” is a prominent topic in the machine learning and NLP communities. Consequently, this research is expected to make a meaningful contribution to the literature.

**Theoretical Claims:**

This paper presents a series of definitions, proofs, and lemmas, most of which appear reasonable. However, I have some concerns, as mentioned above. Additionally, I acknowledge the possibility of mathematical errors in the provided proofs that I may not have detected.

---

> ### Author Rebuttal · Authors · 2025-03-30
>
> We sincerely appreciate the thorough and constructive reviews. Below we address each concern raised by the reviewers.
>
> ## Concerns about the Assumptions
> We note your thoughtful concerns regarding our methodological assumptions, specifically concerning: **(1) the coverage of multiple valid reasoning paths, (2) the impact of error propagation, and (3) the single gold-standard path assumption ($r_l^{*}$) presented in Section 4.** We address each point systematically.
>
> **Response to (1):** We would like to clarify our conceptual framework (and apologize for any lack of clarity in the current manuscript). Our model treats LLM reasoning as a two-phase process: *planning and execution*. Given a question, The planning phase generates implicit thought sequences ($t$ in Fig.1), while the execution phase produces the observable responses ($r$). While we acknowledge multiple paths may lead to correct solutions, our analysis intentionally focuses on error probabilities within individual execution paths. More specifically, though multiple reasoning paths can lead to the answer, **we focus on one specific path and discuss the errors and probabilities on this exact path.**
>
> **Response to (2):** We completely agree that earlier errors typically have greater cumulative impacts. However, our theoretical contribution (Theorem 3.3) specifically examines **the initial error occurrence probability rather than its propagation effects.** This focused analysis provides fundamental insights into the first-error statistics during reasoning chains.
>
> **Response to (3):** Within our execution-phase modeling framework, where reasoning sub-tasks are pre-determined during planning, the gold-standard response assumption ($r_l^{*}$) remains both theoretically sound and practically meaningful for our analytical purposes.
>
> Another possible concern of our setting could be the explanation of the "reflection" mechanism in many modern LLMs, A detailed discussion of its relationship to MI decay appears in our response to Reviewer NREt (Section "The General Linkage between MI Decay and Reasoning Errors"), which we respectfully refer you to for complementary analysis.
>
>
> ## More Experiments
> We appreciate the reviewer's suggestions for expanded experimental validation and are pleased to provide additional empirical evidence or explanations as follows.
>
> 1) **Extra Benchmark Task: Game of 24**.
> We conducted verification experiments using the Game of 24 benchmark ([1]), which requires solving arithmetic puzzles to obtain the number 24. Our results demonstrate consistent patterns in Figure 3:
>
> |  $N$   | 2  | 4 | 6 | 8 | 10 | 12 | 14 | 16 | 18 | 20 |
> |  -  | - | - | - | - | - | - | - | - | -| -|
> | ORM-Vote  | 17.96% | 29.01% | 38.67% | 46.13% | 50.83% | 53.87% | 59.94% | 63.81% | 66.85% | 67.68% |
> | ORM-Max  | 17.96% | 29.01% | 38.67% | 46.13% | 50.83% | 53.87% | 59.94% | 63.81% | 67.13% | 68.23% |
>
> **where $N_{res}=6.08,N_{call}=18.24$, and baseline MCTS obtain an accuracy of 64.80%.**
> This result illustrates a similar conclusion to Fig.3, when the total reasoning cost is comparable, BoN can achieve a comparable and even better performance than MCTS.
>
> 2) **Larger LLMs**.
> Beyond the 7B/8B models in Figure 2, we include additional validation on Qwen2.5-14B-Instruct and Qwen2.5-32B-Instruct. Results are available in our anonymous repository: https://anonymous.4open.science/r/extra-experiments-0E75/README.md, the ***Larger LLMs*** section.
>
> 3) **Alternative Search Algorithms**.
> We categorize existing approaches into:
> - **Non-tree-based methods** (such as AoT): These methods typically incorporate reflection mechanisms that currently **fall outside our theoretical framework focused on width-expansion limitations** in tree-based methods. Alternatively, we discuss potential extensions to model reflection in our response to Reviewer Qdus (section "Extension for more Test-Time Scaling Methods"). We believe this will be an interesting extension of our future work. Thanks for your valuable advice!
> - **Tree-based methods** (BFS, DFS, ToT, etc.): As demonstrated in [1,2], since MCTS are designed as a series of operations including "Selection, Expansion, Simulation and Backpropagation", MCTS has higher probability to select more valuable intermediate thoughts in practice. Thus, for a given total reasoning cost, MCTS usually outperforms naive tree-based methods, as shown in the results of [2]. When it comes to Fig.3, other existing tree-based methods will thus be a baseline lower than MCTS's, and leading to trival results. **Hence, we choose MCTS as the sota baseline of the tree-based methods.**
>
>
> [1] Tree of thoughts: Deliberate problem solving with large language models (NeurIPS 2023).
>
> [2] AlphaZero-like tree-search can guide large language model decoding and training (ICML 2024).
>
> ## Essential References Not Discussed
> We appreciate your suggested references and will incorporate them in our revision.

---

### Official Review · Reviewer_yYwM · 2025-03-22

**Overall Recommendation:** 4

**Summary:**

The paper analyzes the mathematical mechanism behind the slow thinking of large language models. First, the authors define snowball errors using information theory. Then, they derive a lower bound on the probability of reasoning errors based on snowball errors. This bound indicates that the probability of errors increases as snowball errors accumulate. Their further derivation shows that the probability of generating a correct response decreases exponentially with reasoning length. They then prove that expanding the reasoning space through slow thinking can increase the probability of correct reasoning. Finally, they compare two slow-thinking methods, BoN and MCTS, finding that the key factors influencing the results are the capability of the reward function and the total reasoning cost.

**Claims And Evidence:**

Mostly good. All the statements have been theoretically proven. The author provides empirical verification for the existence of snowball errors and the comparison between BoN and MCTS. However, there is no empirical verification for the statement that "the probability of generating a correct response decreases exponentially with reasoning length."

**Essential References Not Discussed:**

As far as I know, no more papers need to be discussed or cited.

**Experimental Designs Or Analyses:**

The selected LLMs are all small (< 10B). The verification will be more robust by selecting models of different sizes.

**Methods And Evaluation Criteria:**

The selected benchmarks are suitable.

**Other Comments Or Suggestions:**

- line 238: leangth -> length.
- Figure 3: set the tick numbers on the x-axis to integers.

**Other Strengths And Weaknesses:**

Strengths: This paper provides a systematic theoretical study of the mechanism of slow-thinking.

Weaknesses: There is no empirical verification for the claim that "the probability of generating a correct response decreases exponentially with the reasoning length L." I believe this verification can be conducted by annotating the reasoning length of each question in a dataset (e.g., GSM8K) and then analyzing the relationship between accuracy and length to determine if accuracy indeed decreases exponentially with the reasoning length.

**Questions For Authors:**

None.

**Relation To Broader Scientific Literature:**

The paper provide a theoretical perspective to analyze the slow-thinking mechanism. `It proves the slow-thinking methods are effective to improve the reasoning correctness of LLMs

**Theoretical Claims:**

Yes, all are correct.

---

> ### Author Rebuttal · Authors · 2025-03-30
>
> We sincerely appreciate your valuable feedback. Below are our point-by-point responses:
>
> ## Lack of Empirical Verification for Proposition 4.3
>
> Proposition 4.3 was designed to facilitate subsequent analyses (Sec 4.2, line 214) while maintaining readability. The negative exponential form provides a simple yet effective characterization of accuracy decay. Below, we address this concern comprehensively by demonstrating **(1) the theoretical robustness of our results under a relaxed assumption** and **(2) empirical validation of the key relationship between accuracy and length**.
>
> **(1) First, we would like to prove that our main results remain valid under a weaker assumption** when $\operatorname{Pr}\left[|\phi(r_l)-\phi(r_l^*)| \leq \tau\right]$ decreases monotonically with $l$.
>
> To address your concern, we relax the original assumption and demonstrate that our core results hold under a more general condition:
> > **Relaxed Proposition 4.3:**
> >
> > Instead of requiring $\operatorname{Pr}\left[|\phi(r_l)-\phi(r_l^*)| \leq \tau\right] = \operatorname{min} \left(\lambda_\tau e^{-l},1 \right)$, we now assume only that the left-hand side *decreases monotonically with $l$ and converges to $0$*, i.e.,
> >
> > $ \operatorname{Pr}\left[|\phi(r_l)-\phi(r_l^*)| \leq \tau\right] = \operatorname{min} \left( \xi(l,\tau),1 \right),$
> >
> > where $\xi(l,\tau) \geq 0$ decreases monotonically with $l$ and converges to $0$.
>
>
> Under this weaker condition, we derive revised bounds presented in the bullet list below:
>
> - **Lemma 4.4:** $\xi^{L}(l,\tau)$,
> - **Lemma 4.5:** $\prod_{l=1}^{L}\epsilon_b \left[ 1-\left( 1-\xi(l,\tau) \right)^k \right]$,
> - **Theorem 4.6:** $\epsilon_b^L k^L \xi^{L}(l,\tau)$,
> - **Lemma 5.1:** $\epsilon_N N^L \xi^{L}(l,\tau)$,
> - **Lemma 5.2:** $\epsilon_b^L b^L \xi^{L}(l,\tau)$,
> - **Lemma 5.3:** $b^{\frac{L(L+1)}{2}}\xi^{L}(l,\tau)\prod_{l=1}^{L}\epsilon_{b^l}$,
>
> where $\xi^{L}(l,\tau) := \prod_{l=1}^{L}\xi(l,\tau)$.
>
> These modifications preserve the validity of Corollary 5.4, Corollary 5.5, and Table 1, confirming that ***our theoretical insights are robust even without the original exponential form in Proposition 4.3***.
>
>
> **(2) Second, we have conducted extra empirical verifications of the relationship between the accuracy and the length.** We provide additional experimental verification (anonymous repository: https://anonymous.4open.science/r/extra-experiments-0E75/README.md, see ***Analysis by Difficulty*** section). The results confirm that **accuracy generally decreases with $l$ within the same difficulty level, exhibiting faster decay in early stages that gradually stabilizes**. We also kindly refer you to our discussion with Reviewer Qdus (section "The Snowball Error and the Length"), which provides additional context on this verification.
>
> ## LLMs in Different Sizes
> We appreciate your suggestion regarding additional experiments with varying model sizes. In response to this valuable feedback, we have conducted extensive analyses on larger language models (Qwen2.5-14B-Instruct and Qwen2.5-32B-Instruct) in addition to the 7B/8B models presented in Figure 2.
>
> The complete experimental results are also available in our anonymous repository: https://anonymous.4open.science/r/extra-experiments-0E75/README.md, we kindly refer you to the ***Larger LLMs*** section.
> We maintained identical experimental settings and workflow as described in Figure 2 to ensure methodological consistency.
>
> Our key findings from these additional experiments demonstrate that:
> - The MI (Mutual Information) decay pattern remains consistent across larger model sizes, exhibiting similar behavior to smaller models.
> - Response quality continues to show a negative correlation with output length, as observed in our original experiments.
>
> These significant findings will be systematically incorporated into our revised manuscript, including comprehensive analysis for the extra results and corresponding updates to Figures/Tables.
>
>
> ## Typos and Figure Ticks
> We thank the reviewer for these careful observations. All typos will be corrected and Fig.3's tick marks will be adjusted to integer values in the final version.
>
> We believe these responses adequately address all concerns raised. We greatly appreciate your time and constructive feedback.

---

> > ### Comment · Reviewer_yYwM · 2025-04-04
> >
> > Thanks for your detailed reply, I will raise my score to 4.

---

> > > ### Author Response · Authors · 2025-04-07
> > >
> > > We are thankful for your generous scoring decision and insightful comments, which helped improve our paper.

---

### Decision · Program_Chairs · 2025-05-01

**Decision:**

Accept (poster)

**Comment:**

This paper presents a timely and theoretically grounded analysis of slow-thinking mechanisms in LLMs through the lens of information theory, introducing the notion of snowball errors to formalize error accumulation in reasoning. Reviewers note that some theoretical assumptions may oversimplify real-world dynamics or lead to loose bounds. Although some key claims, such as exponential decay of correctness, lack direct experimental confirmation, reviewers generally agree on the novelty and clarity of the contribution. Despite the raised concerns, the paper opens a promising direction for better understanding and designing inference-time reasoning strategies. Hence, I decide to recommend for weak accept.